# Harnessing large-language models to generate private synthetic text

## Abstract

Differentially private training algorithms like DP-SGD protect sensitive training data by ensuring that trained models do not reveal private information. An alternative approach, which this paper studies, is to use a sensitive dataset to generate synthetic data that is differentially private with respect to the original data, and then non-privately training a model on the synthetic data. Doing so has several advantages: synthetic data can be reused for other tasks (including for hyper parameter tuning), retained indefinitely, and shared with third parties without sacrificing privacy.

However, generating private synthetic data is much harder than training a private model. To improve performance on text data, recent work has utilized public data by starting with a pre-trained generative language model and privately fine-tuning it on sensitive data. This model can be used to sample a DP synthetic dataset. While this strategy seems straightforward, executing it has proven problematic. Previous approaches either show significant performance loss, or have, as we show, critical design flaws.

In this paper we demonstrate that a proper training objective along with tuning fewer parameters results in excellent DP synthetic data quality. Our approach is competitive with direct DP-training of downstream classifiers in terms of performance on downstream tasks. Further, we demonstrate that our DP synthetic data is not only useful for downstream classifier training, but also to tune those same models.

## 1 Introduction

Machine learning models can memorize their training data (Carlini et al., 2019) and it is possible to extract the training data from a model (Carlini et al., 2021). Training a model with differential privacy (DP) (Abadi et al., 2016) provably reduces the risk of memorization (Ponomareva et al., 2022), which is critical when ML models are trained on sensitive data. However, DP training only ensures that *the model* does not release private information, and just releasing the model or its predictions is not adequate for many applications. For example, other researchers might want to use the data for analysis, or to build a different predictive model. It would therefore be ideal to release the dataset itself while protecting the privacy of the users that contributed to it.

*Local differential privacy* has been proposed as a method of preprocessing low-dimensional datasets before public release (Ponomareva et al., 2023). Local DP adds noise to individual data points in the training data. While protecting privacy, local DP generally leads to much lower utility, due to the large amount of noise that must be added compared to *central differential privacy*, where DP is applied to the model or statistical output (Wang et al., 2017; Bassily et al., 2017; Team, 2017). Generally there is an the inherent tension between *privacy* and *utility* when releasing private datasets: we want to release a dataset that protects the privacy of the underlying data while at the same time we want the dataset to be as useful as the original data for *any* possible downstream task. Therefore, we focus on central DP and consider generating private synthetic data. Generating such synthetic data involves creating a generative model that learns the original data distribution. To protect the original data, either the generative model should be made private, via DP training, or privacy should be enforced at inference time (e.g., during the generation of synthetic data items, so-called private prediction). Private inference has been shown to be inferior to DP training when a large number of

inferences is required (van der Maaten & Hannun, 2020). Since we seek to generate at least as much data as in the original dataset, DP training is the clear choice.

Several works proposed using publicly pre-trained large language models (LLM) for private synthetic data generation (Bommasani et al., 2019; Yue et al., 2022; Putta et al., 2023; Mattern et al., 2022). This approach involves privately fine-tuning an LLM using class labels as prompts for the model and subsequently sampling from this model. However these attempts have had mixed success: they either reported poor utility even for non-private synthetic data, or had to augment standard NLP loss metrics to assist the LLM to correctly respond to prompts during the generation process. Additionally, none of the previous work considered privacy leakage from a pre-trained LLM itself. The privacy leakage happens because these papers used academic datasets (like IMDB (Maas et al., 2011)) as sensitive dataset and they utilized GPT-2 LLM (Radford et al., 2019) which was pre-trained on these datasets without any privacy guarantees.

Although we follow a similar recipe *conceptually*, in that we use a DP-finetuned LLM model to generate private synthetic data, we highlight the following differences in our execution of this idea:

1. *Privacy leakage mitigation*. We draw attention to the need to account for the data that went into pre-training of the LLMs used for generation. Our de-duplication of the pre-training data ensures that no privacy leakage, possibly present in previous works, takes place.
2. *Reporting*: We use a long sequence of text (512 tokens, representing full reviews like IMDB or Yelp) as our privacy unit. Our privacy guarantees (Appendix A) are tight and transparent, and we tune the hyperparameters of the downstream classifier on private synthetic data only.
3. *Method*: We demonstrate that the standard approach to private fine-tuning does not yield the desired quality of generated data. Instead of augmenting the LLM's objective or architecture for fine-tuning as in (Putta et al., 2023; Mattern et al., 2022), we identify a loss function, well known to the NLP community, that is particularly suitable for private fine-tuning. Additionally, we argue that parameter-efficient fine-tuning, especially LoRA tuning, is beneficial for synthetic data generation

Our contributions can be summarized as follows:

1. We demonstrate state-of-the-art results in terms of quality of synthetic data. Specifically, we show in multiple experiments that the quality of the model trained on private synthetic data is comparable to or even better than the quality of the downstream model trained on real data with DP.
2. We demonstrate that parameter efficient fine-tuning like prompt-tuning and LoRA-tuning is superior to full fine-tuning when the tuning is performed privately. In particular, LoRA-tuning results in up to 11 percentage points lift in downstream model performance. To the best of our knowledge, we are are the first to demonstrate that parameter-efficient tuning performs better than full fine-tuning when each is combined with DP, whereas the opposite often holds for non-DP training (Shin et al., 2020; Brown et al., 2020; Zhong et al., 2021).
3. We show that generating more synthetic data than the size of the original dataset is helpful, especially for simpler downstream models.
4. We show that DP synthetic data can be used to tune the hyperparameters of the downstream classifiers. We achieve ranking correlation with the ordering of trials performed on real data of up to 87%, even for $\epsilon = 1$.

## 2 RELATED WORK

Privacy-preserving synthetic data generation requires that the generated data is both *high-fidelity* (*i.e.*, exhibits similar distributional characteristics as the original data) and *anonymized* to preserve the privacy of the users who contributed their data. For complex data like text, images, audio and video, most existing approaches build a generative model, for example a GAN-based model (Guan et al., 2018). However, in most previous work the data is anonymized using heuristic methods, without providing formal privacy guarantees. For example, Melamud & Shivade (2019) attempted to de-identify summaries of clinical discharge notes using heuristic rules for an LSTM model and only empirically demonstrated the privacy of the synthetic data.

DP-fine tuning is a standard method for fine tuning LLMs that satisfies differential privacy guarantees and has been shown to perform well with appropriate hyperparameter tuning (Li et al., 2021; Yu et al.,

2021). DP-fine tuning involves using a pre-trained model and a modification of a training algorithm like DP-SGD to fine tune the model on private data.

For private synthetic text generation, Bommasani et al. (2019) suggested using a pre-trained GPT-2 model and then DP-fine tuning it on private data with word-level privacy, but did not implement or evaluate any method. In similar vein, Yue et al. (2022) DP-fine tuned pre-trained GPT models of various sizes. While they do obtain good results on some of the benchmarks, they also observe up to 25% drop of downstream model accuracy on synthetic data (even without DP) on other benchmarks. Putta et al. (2023) attempted a similar recipe on a pre-trained distilGPT2 model, but also reported a large performance drop of the classifier trained on synthetic data. Additionally, they proposed modifying the fine tuning process to also include a discriminator that attempts to distinguish between the labels to improve the separability of learned representations for two binary classes of the text data. Similarly, Mattern et al. (2022) proposed augmenting the training objective with an additional term penalizing the generation of sample with the wrong label.

None of the prior work takes into account problem of data contamination between LLM pre-training dataset and dataset used in downstream task. As we show in Appendix D this problem is real. In particular some of both training and test samples examples from downstream datasets could be found in GPT-2 pre-training data, which is used by all prior work. This may potentially invalidate DP-guarantees and may result in overestimated accuracy on downstream tasks.

Additionally none of the works *on DP synthetic data* mentioned above explored parameter-efficient fine tuning. To the best of our knowledge, we are the first to demonstrate that parameter-efficient finetuning like LoRA tuning can produce better quality synthetic DP data than full finetuning.

## 3 PRELIMINARIES

**Differential privacy**    Differential Privacy (DP) (Dwork et al., 2006b) is considered the gold standard for ensuring data anonymization. Throughout this work we employ a notion of DP called $(\epsilon, \delta)$-DP.

**Definition 1** ($(\epsilon, \delta)$-Differential Privacy, (Dwork et al., 2006a))**.** *Consider neighbouring datasets to be datasets that differ only in addition or removal of one record only. Given non-negative $\epsilon$ and $\delta \leq 1$, a mechanism $\mathcal{A}$ is $(\epsilon, \delta)$-DP if for any two neighbouring datasets $D$ and $D'$ and for any $S \subseteq Range(\mathcal{A})$,*

$$P[\mathcal{A}(D) \in S] \leq \exp(\epsilon) \times P[\mathcal{A}(D') \in S] + \delta. \tag{1}$$

The $\epsilon$ and $\delta$ values determine the strength of the privacy guarantees, with smaller values corresponding to stronger guarantees. The *post-processing* property of a DP mechanism means that applying any data-independent transformation to its output will remain DP with the same guarantees.

**DP in context of ML models**    In context of ML, DP can be introduced either at the input level, during the training of a model (DP-Training), or during model serving (prediction) (Ponomareva et al., 2023). DP synthetic data falls into the first category and in general is a harder task than introducing DP during the training. This is because DP synthetic data ensures that *any* ML model trained on this data is DP with respect to the original training data. This is in contrast with DP-Training that only ensures that a particular ML model is DP. Therefore, it is expected that any model trained on DP synthetic data should perform *at most* as well as the downstream DP-Trained ML model on real data. However the idea of using a pre-trained generative LLM to aid generation of synthetic data means that we inject *massive amount* of public data, making the task of DP synthetic data generation less daunting.

The most practical methods of DP-Training for non convex losses are gradient-noise injection methods like DP-SGD (Abadi et al., 2016), which work by clipping per example gradients to limit the sensitivity of the loss, and noising aggregated clipped gradients with Gaussian noise to make them private. The noise level is proportional to the clipping norm (the sensitivity) and the strength of $\epsilon$ guarantees. The same recipe can be adopted to adaptive optimizers like Adafactor (Shazeer & Stern, 2018), where the noised gradients are passed to the optimizer to figure out the optimal learning rate.

**LLMs**    Throughout the paper we will use the terms of pre-training and fine-tuning of LLMs: pre-training is the initial training of a LLM with a large public dataset, for example C4 (Raffel et al.,

2019). Fine-tuning is an adaptation of a pre-trained model to perform some concrete task, for example question-answering, which involves running several epochs of an optimizer over the additional task training data.

# 4 METHODOLOGY

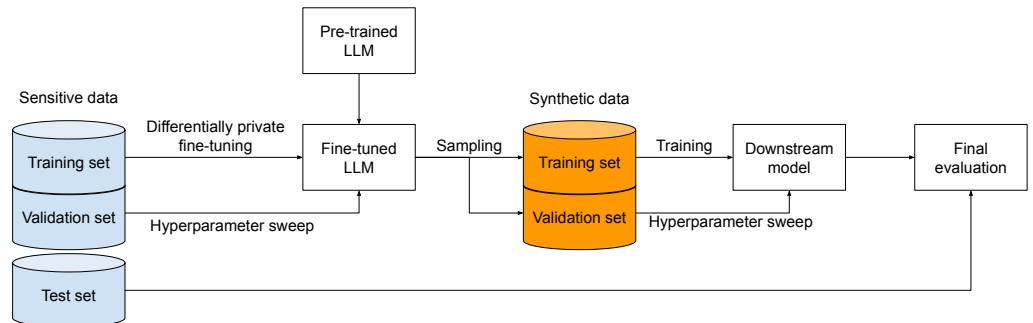

Figure 1: Synthetic data generation and evaluation.

As a motivational example, consider the task of medical data sharing for research purposes: a medical provider has a sensitive dataset with patients records and wants to accomplish some machine learning task. They may want to share the dataset with external researchers and academic institutions to get their help in solving the downstream task, while preserving the privacy of the original data.

We assume that we have a **sensitive dataset** $D$ consisting of $(D_{train}, D_{valid}, D_{test})$, where the privacy of each record must be protected (see additional details on the unit of privacy in Appendix A). We want to accomplish some task on this dataset, such as training some **downstream** machine learning **model**. Additionally, we would like to allow a non-trusted third party to be able to perform the downstream task without violating privacy. To achieve this, we aim to create a **synthetic dataset** $D^{synth}$, which is DP with respect to the dataset $D$. Our dataset $D^{synth}$ will consist of synthetic training and validation splits. Figure 1 illustrates our methodology of data generation and evaluation:

1. Privately finetune (e.g., using DP-Training) a publicly pre-trained generative LLM $G$ on $D_{train}$, using $D_{valid}$ for hyperparameter tuning. To tune hyperparameters for DP-Training, we follow an algorithm outlined in (Ponomareva et al., 2023) (Section 5.4.1).
2. Independently sample G to generate two new synthetic datasets $D_{train}^{synth}$ and $D_{valid}^{synth}$ which will serve as synthetic training and validation data.
3. Train a downstream model $M$ on $D_{train}^{synth}$ and use $D_{valid}^{synth}$ for hyperparameter tuning.
4. Evaluate the final performance of the model on real dataset $D_{test}$.

## 4.1 USING AN LLM FOR DATA SYNTHESIS

Both encoder-decoder or decoder-only pretrained language models can generate synthetic data; we use decoder-only LLMs in our experiments. To finetune LLM for the synthetic data generation task, we use the next token prediction objective setup as follows. Given an example from the sensitive dataset with text $x$ and label $y$, we generate a prefix $p = $ "`[TaskName] [LabelName`$_y$`] `", where "`[TaskName]`" is the name of the task (for example "`[imdb]`"), and "`[LabelName`$_y$`]`" is "`[negative]`" when $y = 0$ or "`[positive]`" when $y = 1$. We finetune the model using the Prefix-LM objective (Raffel et al., 2020) using $p$ as a model input and $x$ as a target.

Below we outline how the Prefix-LM way of splitting of the training example into input and target is advantageous for DP-training. Let's consider some example from the dataset which is tokenized into input prefix $p = \{z_1, \ldots, z_k\}$ and target $x = \{z_{k+1}, \ldots, z_n\}$. Typically, weighted next token prediction cross entropy loss looks like the following: $L(\vec{z}, \vec{w}, \theta) = -\sum_{i=1}^{n} w_i z_i \log P(z|z_{<i}, \theta)$ where $\theta$ - model parameters, $\vec{z} = \{z_1, \ldots, z_n\}$ is tokenized training example (including input and target tokens), each $z_i$ as one hot encodings of token, $P(z|z_{<i})$ is the probability of $i$-th token given values $z_{<i}$ of all previous tokens and $\vec{w} = \{w_1, \ldots, w_n\}$ is weights vector for each token in the loss.

Standard next-token prediction loss assigns weights $w_i = 1$ to all tokens, including those in the prefix $p$. As a result prefix tokens will be included in the gradient of the loss $\frac{\partial L}{\partial \theta}$, thus essentially forcing the model to learn the distribution of tokens in the prefix as well. On the other hand, the Prefix-LM formulation assigns zero weights to the prefix[1], i.e. $\forall i \leq k : w_i = 0$, so the total loss looks like the following: $L_{\text{PrefixLM}}(\vec{z}, \vec{w}, \theta) = -\sum_{i=k+1}^{n} z_i \log P(z|z_{<i}, \theta)$

As a result the LLM is not forced to learn distribution of input prefix $p$ which we found to be beneficial for differentially-private training. DP-Training adds the noise to all the gradients, in a standard setup this will result in the gradients from the prefix portion being corrupted with the noise. This in turn means that prompting the DP-Trained LLM to generate synthetic data will not work as well as expected. We believe this is the same phenomenon that was observed in works Putta et al. (2023) and Mattern et al. (2022), where authors had to add an adversarial head or augment the loss respectively, to aid the model in differentiating different types of prompts. Prefix-LM in turn is a standard loss well known to the community, and this comes with the benefits of knowing approximate hyperparameter values for its tuning. The aforementioned Prefix-LM setup allows to train one model for all the class labels and can be easily extended beyond the binary classification setup.

### 4.2 PARAMETER-EFFICIENT FINE TUNING

Full finetuning of large models is expensive, and, empirically, tuning very large number of weights with DP-finetuning often results in substantial utility drop. Many techniques exist that update the pretraining model without resorting to full model weights update. In this work, we consider two popular ones - Prompt Tuning and LoRA.

**Prompt tuning** (Lester et al., 2021) is a technique which prepends a small *prompt tensor* in front of the model's input in the embedding space, freezes the rest of the model's parameters and then finetunes only the prompt tensor weights. We found that combining prompt tuning with differentially private training allows us to achieve much higher utility of the trained generative model compared to full model fine-tuning. This could be explained by the fact that the prompt tensor is much smaller compared to the size of entire model (we used prompt tensor with 20480 parameters vs 8B weights in the full model) and smaller models tend to have smaller gap between private and non-private utility (Bassily et al., 2014; Bun et al., 2014), probably due to the total amount of noise injected during the training. It should be noted that prompt tuning as described in original paper (Lester et al., 2021) showed very poor utility when trained with differential privacy. We observed that even in the best runs LLM quality metrics (perplexity, next token prediction accuracy) fluctuated significantly. No amount of typical hyperparameter tuning could improve prompt tuning utility in DP-regime.

Borrowing some ideas from (Mehta et al., 2022) and experimenting with various optimizers and ways to initialize prompt tensor proved to be the key to making prompt-tuning work. Eventually we found out that the main culprit of poor utility was prompt tensor initialization. (Lester et al., 2021) initializes prompt tensor by using embeddings of some real tokens from vocabulary. Changing prompt tensor initialization to random uniform with small range $[-0.01, 0.01]$ significantly improved utility. Additionally we observed that change of optimizer from Adafactor to Adam or Momentum helped to make training more stable, which simplified hyperparameter tuning (Appendix E).

**LoRA tuning** (Hu et al., 2021) (Low-rank Adaptation) is a technique that freezes all the pre-trained model weights and introduces trainable low-rank decomposition matrices into each dense layer (MLP and Attention). This results in fewer trainable weights than full fine tuning but the number of trainable weights in LoRA is significantly larger than in Prompt tuning. For example, rank 8 LoRA updates 20M trainable parameters, as opposed to 41K prompt tuning vs 8B full fine tuning. Empirically we find (Section 5) that LoRA results in superior performance, surpassing that of both full finetuning and Prompt finetuning and that tuning both MLP layers and Attention blocks is preferred, see Appendices F and J.5 for more details.

As a conclusion, we advocate for the use of parameter-efficient techniques when performing DP-training, with LoRA being the most promising so far.

---

[1]Original paper (Raffel et al., 2020) only describes bidirectional attention over prefix and omits the description of loss weights. Nevertheless zero weighting of the prefix is implemented in the T5 code.

## 4.3 Data sampling

To generate one synthetic example we first randomly select example label $y$, create a prefix $p =$ `"[TaskName] [LabelName_y] "` (Section 4.1), feed prefix $p$ as an input to the language model, and autoregressively sample the output. We repeat this process many times until we reach the desired amount of synthetic data. For each task we sampled at least the same amount of synthetic data as in original training dataset. We observed that generating more synthetic examples generally improves downstream task performance, but this benefit eventually diminishes and compute is typically the limiting factor (Appendix G).

## 5 Experiments

**Generative LLM** In our experiments we used a model with architecture similar to Lamda 8B (Thoppilan et al., 2022), which we pre-trained on The Pile dataset (Gao et al., 2020) using a standard next-token prediction loss. We stress that for our experimental results to be valid we must ensure that the pre-trained model was not itself trained on data that is considered private for the downstream task. For example, the GPT-2 model used in (Mattern et al., 2022) seemingly contained IMDB data in its pre-training dataset (Radford et al., 2019), but this model was subsequently used to generate a synthetic version of IMDB, see also appendix D for details. To prevent privacy leakage we modified the pre-training dataset by de-duplicating it against all sensitive datasets used in downstream tasks, following the recipe and scripts from (Lee et al., 2022). The outline of the de-duplication approach is as follows. First we tokenized and constructed a suffix array for each involved dataset (The Pile, IMDB, Yelp, AGNews). Then we used the suffix arrays to find common sequences of 50 or more tokens which appear in The Pile and any other dataset. Finally we cut all those common sequences from The Pile dataset. Note that this de-duplication is "stronger" than simply removing the datasets from the Pile. After cutting the sequences we de-tokenized the dataset back to strings and used it for pre-training. Refer to Appendix C for additional details.

**Datasets and classification problems** We conducted our experiments on IMDB (Maas et al., 2011), Yelp (Zhang et al., 2015a) and AGNews (Zhang et al., 2015b) datasets. All these datasets only provide a training and test set, so in each case we use the first 90% of the training set for training and the remaining 10% for validation. For each dataset we formulated a binary classification problem (sentiment classification) as the downstream prediction task.

## 5.1 Downstream classifier performance

We investigate the utility of using private synthetic data for a downstream task. For each dataset, we consider two types of models. First one is a (encoder-only) BERT model (Devlin et al., 2018a) with classification head. BERT is publicly pretrained and then fine tuned using either real data or our generated synthetic data. This model benefits from public pre-training data. We also consider a word-level CNN model (Johnson & Zhang, 2015) that does not utilize any public data. For each model, we report the performance on real data with no DP guarantees (an entry *"Real"* with $\epsilon = \infty$ in Table 1). This serves as a upper bound of downstream classifier performance. We also report the performance of doing DP-Training on the downstream classifier directly (entries *"Real"* with $\epsilon \in (1, 3, 10)$, referred to as *"DP-on-real"* in the text) and report the results on synthetic data generated from fine-tuned (*Fine-tuned-SD*), prompt tuned (*Prompt-tuned-SD*) and LoRA-Tuned (*LoRA-tuned-SD*) models. We would like to highlight however that in the case of using the real data directly for DP-Training, only the resulting downstream model is DP, and the real data can't be shared freely or used for hyperparameter tuning (or such tuning should be accounted for in privacy guarantees). DP Synthetic data however can be shared freely and used for feature engineering, hyperparameter tuning, etc.

**Non-private synthetic data** Firstly, our results in Table 1 indicate that obtaining good fidelity non-private synthetic data is possible, contrary to the results reported in (Yue et al., 2022) and Putta et al. (2023). Both Fine-tuned-SD and LoRA-tuned SD exhibits better performance than Prompt-tuned-SD, in line with current understanding that for a non-DP setting, tuning more model parameters is beneficial (Shin et al., 2020; Brown et al., 2020; Zhong et al., 2021). Interestingly, even for non DP setting, downstream models trained on LoRA synthetic data outperform those trained on fully fine-tuned synthetic data in 2 out of 3 datasets.

**Private synthetic data**    While there is a clear utility drop when going from non-private SD data to private SD, DP LoRA-tuned-SD is a clearly superior way of obtaining DP synthetic data. Prompt-tuned DP SD is better than fully fine tuned DP SD, however LoRA outperforms the Prompt-tuned DP synthetic data in majority of the cases. We hypothesize that it might be due to less total noise being added in DP LoRA models, due to fewer parameters being updated than with the full fine-tuning. Prompt tuning on the other hand updates the minimal number of parameters, however this minimum update hurts the utility of SD, suggesting that like with everything in ML, there is a "sweet spot" on the number of parameters trained with DP.

The difference between the performance is significant, with LoRA-tuned-SD exhibiting of up to 10-11% lift on downstream BERT classifier tasks, compared to model trained on fine-tuned-SD. For CNN model that is more dependent on the quality of the data than BERT (that essentially reaps some additional benefits from Transfer Learning), the results are even more significant, with a boost from prompt-tuned-SD (vs fine-tuned-SD) reaching up to 22%.

**Private synthetic data vs DP-Training on real data**    To obtain a DP downstream ML model, we can either use DP synthetic training data or introduce DP directly during downstream model training (*DP-on-real*). As previously mentioned, the former is a harder setup. When comparing BERT models, we can see that private LoRA-tuned-SD achieves performance similar **or even superior** (e.g., for IMDB and Yelp datasets) to DP-on-real for all levels of privacy, but an additional benefit of such synthetic data is that it can be additionally shared freely and used for hyperperamter tuning and feature engineering. For CNN model, LoRA-tuned-SD (and even prompt-tuned SD) exhibits **better** performance than DP-on-real. This is due to the fact that private synthetic data benefits from massive amount of public data that was used for pretraining of the LLM (CNN model itself is trained from scratch, as opposed to BERT that is itself a pre-trained model, albeit with smaller amount of public data than the 8b Lamda model we used for SD generation). This indicates that for simpler models synthetic data can be a preferred way of injecting additional public knowledge. This is an interesting result since it is commonly assumed that for Transfer Learning to work, public data should come from a similar distribution as the target data. However in case of synthetic data, we inject public data from different distributions (crawl of the web) than that of the downstream task (e.g. Yelp reviews).

Table 1: Accuracy of BERT and CNN downstream classifiers on IMDB, Yelp and AGNews datasets.

| | $\epsilon$ | BERT | | | | CNN | | | |
| | | Real | Synthetic | | | Real | Synthetic | | |
| | | | Finetune | Prompt-tune | LoRA | | Finetune | Prompt-tune | LoRA |
|---|---|---|---|---|---|---|---|---|---|
| IMDB | $\infty$ | **93.7 ± 0.1** | 93.2 ± 0.2 | 92.0 ± 0.1 | 91.6 ± 0.2 | **90.1 ± 0.1** | 89.8 ± 0.1 | 87.4 ± 0.1 | 89.0 ± 0.1 |
| | 10 | 90.6 ± 0.1 | 84.0 ± 0.7 | 90.7 ± 0.2 | **91.3 ± 0.2** | 78.2 ± 0.4 | 80.0 ± 0.5 | 86.9 ± 0.1 | **87.7 ± 0.2** |
| | 3 | 89.7 ± 0.2 | 83.9 ± 0.6 | 87.4 ± 0.2 | **90.6 ± 0.2** | 74.8 ± 0.6 | 74.2 ± 0.1 | 85.4 ± 0.5 | **87.4 ± 0.3** |
| | 1 | 88.6 ± 0.1 | 79.1 ± 1.7 | 88.1 ± 0.4 | **90.0 ± 0.3** | 69.3 ± 0.6 | 64.7 ± 0.5 | 85.4 ± 0.1 | **87.6 ± 0.4** |
| Yelp | $\infty$ | **97.6 ± 0.1** | 95.9 ± 0.1 | 93.9 ± 0.1 | 96.4 ± 0.1 | **95.6 ± 0.1** | 89.3 ± 0.3 | 91.6 ± 0.1 | 93.7 ± 0.0 |
| | 10 | 94.9 ± 0.1 | 84.2 ± 0.3 | 94.1 ± 0.1 | **95.9 ± 0.1** | 91.8 ± 0.1 | 71.6 ± 1.5 | 91.0 ± 0.4 | **93.9 ± 0.1** |
| | 3 | 94.6 ± 0.1 | 84.6 ± 0.1 | 93.5 ± 0.1 | **95.6 ± 0.1** | 90.9 ± 0.2 | 67.9 ± 2.6 | 90.5 ± 0.1 | **93.6 ± 0.1** |
| | 1 | 94.3 ± 0.1 | 84.1 ± 0.3 | 94.1 ± 0.1 | **95.5 ± 0.1** | 89.8 ± 0.1 | 71.1 ± 0.4 | 91.1 ± 0.3 | **93.4 ± 0.1** |
| AGNews | $\infty$ | **93.7 ± 0.1** | 91.1 ± 0.1 | 88.3 ± 0.3 | 91.8 ± 0.2 | **91.3 ± 0.1** | 87.7 ± 0.1 | 84.7 ± 0.1 | 88.5 ± 0.2 |
| | 10 | **90.9 ± 0.2** | 65.1 ± 5.3 | 86.9 ± 0.1 | 90.0 ± 0.1 | 85.2 ± 0.2 | 45.2 ± 1.3 | 83.5 ± 0.2 | **86.9 ± 0.1** |
| | 3 | **90.5 ± 0.1** | 65.3 ± 2.7 | 86.2 ± 0.2 | 89.6 ± 0.1 | 83.2 ± 0.1 | 45.8 ± 2.1 | 83.2 ± 0.1 | **86.3 ± 0.2** |
| | 1 | **89.8 ± 0.2** | 65.7 ± 2.9 | 83.9 ± 0.8 | 89.4 ± 0.1 | 79.9 ± 0.2 | 46.8 ± 1.5 | 80.4 ± 0.6 | **85.8 ± 0.1** |

**Amount of synthetic data vs downstream classifier performance**    We studied how much synthetic data we should generate relative to amount of real data. Table 2 demonstrates that generating more synthetic data can be beneficial, but has diminishing returns for BERT (0.8% lift going from 1x to 3x times the data), with benefits more pronounced for simple models like WordCNN (1.4% lift from increasing the amount of synthetic data 3x).

Table 2: Amount of synthetic data vs downstream classifier performance for IMDB prompt-tuning, $\epsilon = 1$

| Model | 1x | 2x | 3x | 4x | 5x | 6x |
|---|---|---|---|---|---|---|
| BERT | 87.2 ± 0.4 | 87.9 ± 0.4 | 88.0 ± 0.1 | 88.1 ± 0.4 | 88.4 ± 0.1 | 88.7 ± 0.1 |
| WordCNN | 83.2 ± 0.2 | 84.3 ± 0.4 | 84.6 ± 0.1 | 85.4 ± 0.1 | 85.7 ± 0.3 | 85.8 ± 0.2 |

One can also potentially combine the synthetic data with training with DP on real data, by pre-training the downstream model with DP synthetic data and then fine-tuning with DP on real data. This will

however require spreading the privacy budget between DP synthetic data and DP-Training of the downstream classifier. We leave this for future work.

**Comparison with prior work**  While works below don't provide sufficient (or any) information on their privacy unit (as we do in Appendix A), we assume that privacy unit that was used is one example (e.g. 1 full yelp or imdb review etc); we also assume central DP setting, that $\delta$ values are the same or comparable etc. Additionally, none of the works below take into account the fact that pre-training data might have contained the data they deem private (as we highlight in Appendix D), *potentially invalidating their reported DP guarantees.*

Yue et al. (2022) used Yelp dataset for multiclass (rating) classification, so our results are not directly comparable. Putta et al. (2023) used AGNews dataset. Their approach is a combination of next token prediction (similar to our setup) and additional loss term from a new head that attempts to learn to distinguish between various classes directly (instead of simply relying on the prompts in the next token prediciton head). Putta et al. (2023) reports 0.867 accuracy of downstream task for $\epsilon$ of 3, while we obtain 89.6 (the baseline performance of downstream classifier for our and their work is comparable, 0.938, suggesting that we are using comparable downstream classifiers). Mattern et al. (2022) suggested a modification of the loss (prompt-mismatch loss, to discourage the generation of text inconsistent with the prompt, like generating a negative review when positive prompt was given). They performed experiments on IMDB dataset. Their best IMDB experiments reporting worse accuracy on DP synthetic data (89.1% theirs vs 90.6% ours for $\epsilon = 3$). They also seem to have worse performance on real data despite using the same model (BERT-classifier).

## 5.2 Tuning downstream model hyperparameters on synthetic data

With the following experiments on IMDB data, we want to demonstrate that private synthetic data is useful for hyperparameter tuning of the downstream classifier. For all our experiments, when tuning the downstream classifier, we use validation accuracy on set-aside portion of synthetic data for hyperparameter selection. We tune weight decay and learning rate for both CNN and BERT models. For synthetic data, we create vectors of accuracy on validation (synthetic) data and performance on real test data for all possible combinations of hyperparameter values tried. We then report the ranking correlation between performance as indicated by validation accuracy (synthetic data) and test accuracy computed on real data. We also report the ranking correlation of accuracies on real validation and real test data, to provide an upper bound. Additionally, we report rank-biased overlap ranking metric (Webber et al., 2010), which is a weighted metric that gives more weight to the top of the ranking (we use parameters that give 85% of the weight to the first top 25% of ranking).

Table 3 demonstrates excellent ranking correlation on synthetic data. Interestingly, prompt-tuned synthetic data metrics, in particular the mean and standard deviation of the top 25% of trials, suggest that BERT classifier performance is less sensitive to hyperparameters on better fidelity (prompt or LoRA tuning) data than on worse fidelity data (fine-tuning).

## 5.3 Estimating synthetic data quality

It is useful to have an efficient method of evaluating the quality of a synthetic dataset without relying on specific downstream tasks. For one, a key use case for privacy preserving synthetic data is to enable data sharing without a definitive end use case. For another, training the generator LLM has multiple hyperparameters that can be tuned, and it can be prohibitive to evaluate candidate models using full data synthesis and downstream training (which itself might require tuning hyperparameters). Instead, lighter weight proxy metrics can be used. Commonly used proxy metrics are: perplexity, n-gram statistics, and MAUVE (Pillutla et al., 2021). We investigate the effectiveness of each of these metrics by comparing their correlation to downstream performance (Table 4). These metrics are used to compare datasets, and thus their absolute value is uninformative. For n-gram statistics we determine the frequency of unigrams, bigrams, and sample lengths in characters for both the original and synthetic datasets. We then compute the area under the divergence frontier between these two frequency distributions as is done by MAUVE. MAUVE works by computing the difference between two datasets by first embedding each example, then clustering the datasets, followed by comparing (via divergence frontiers) the histogram of cluster membership across the two datasets. It has recently been shown to be an effective metric for synthetic text datasets (Yue et al., 2022) (Mattern

Table 3: Ranking correlations (full list) and rank-biased overlap (RBO) (Webber et al., 2010) for top 25% of hyperparameter tuning trials. Real data metrics are calculated on the performance of a model as reported on real validation and real test. For synthetic data, metrics are calculated on synthetic validation and real test data. *Mean 25%* and *STD 25%* show mean and std of real test accuracy evaluated on top 25% trials (ordered by validation accuracy on synthetic data).

| Model | $\epsilon$ | Method | RBO 25% | Spearman | Kendall | Mean 25% | STD 25% |
|---|---|---|---|---|---|---|---|
| BERT | $\infty$ | Real data | 0.56 | 0.96 | 0.93 | 93.55 | 0.50 |
| | 1 | | 0.33 | 0.94 | 0.86 | 79.27 | 0.75 |
| | 3 | Fine-tuning | 0.59 | 0.93 | 0.85 | 84.09 | 0.29 |
| | 10 | | 0.33 | 0.83 | 0.71 | 84.91 | 0.67 |
| | 1 | | 0.32 | 0.73 | 0.60 | 88.00 | 0.00 |
| | 3 | Prompt-tuning | 0.22 | 0.68 | 0.54 | 87.18 | 0.39 |
| | 10 | | 0.31 | 0.75 | 0.61 | 90.27 | 0.45 |
| | 1 | | 0.29 | 0.86 | 0.79 | 90.00 | 0.00 |
| | 3 | LoRA-tuning | 0.23 | 0.7 | 0.56 | 90.5 | 0.5 |
| | 10 | | 0.3 | 0.78 | 0.66 | 91.18 | 0.39 |
| WordCNN | $\infty$ | Real data | 0.92 | 0.92 | 0.84 | 90.00 | 0.00 |
| | 1 | | 0.80 | 0.87 | 0.77 | 64.00 | 2.59 |
| | 3 | Fine-tuning | 0.63 | 0.79 | 0.65 | 72.09 | 3.87 |
| | 10 | | 0.40 | 0.76 | 0.62 | 78.45 | 2.46 |
| | 1 | | 0.64 | 0.73 | 0.59 | 84.36 | 1.49 |
| | 3 | Prompt-tuning | 0.73 | 0.77 | 0.63 | 84.82 | 1.40 |
| | 10 | | 0.76 | 0.80 | 0.69 | 86.27 | 1.05 |
| | 1 | | 0.92 | 0.87 | 0.78 | 88.00 | 0.00 |
| | 3 | LoRA-tuning | 0.69 | 0.81 | 0.67 | 87.45 | 0.66 |
| | 10 | | 0.78 | 0.84 | 0.75 | 87.82 | 0.57 |

et al., 2022) (Kour et al., 2022), which our results support. We compute the MAUVE score as given in Pillutla et al. (2021) using the suggested hyperparameters unless noted. We investigated modifying these hyperparameters and confirm they make little difference to the relative ranking, with the notable exception of the model used to embed examples. Unlike the original paper, we find larger models to be much more effective. In particular, embedding using Sentence-T5 (Ni et al., 2021) has much higher correlation to downstream performance than BERT or any other model we tried. For more details see appendix K. Our results match many of the results given in Kour et al. (2022). All metrics are at least somewhat noisy with standard test-set perplexity performing very well. Given its ease to compute while finetuning, perplexity is our recommended proxy metric when available.

Table 4: The Spearman's rank correlation for each metric compared against downstream classifier performance. Metrics are used to select candidate datasets, and thus their relative rank is whats most important for the metrics to reflect.

| Perplexity | Unigram | Bigram | Length | MAUVE | | |
|---|---|---|---|---|---|---|
| | | | | BERT | St5-base | St5-3B |
| $0.91 \pm 0.02$ | $0.74 \pm 0.11$ | $0.83 \pm 0.09$ | $0.88 \pm 0.26$ | $0.84 \pm 0.62$ | $0.88 \pm 0.04$ | $0.93 \pm 0.10$ |

# 6 CONCLUSION

We have shown that training downstream models on DP synthetic training data is an effective alternative to training such models with DP directly on real data for text classification tasks. We explored two methods for privately generating the synthetic training data, both of which involve modifying the weights of an existing LLM. One method privately fine-tuned all the layers of the LLM, while the other method used parameter efficient fine tuning ( 'prompt-tuning' and 'LoRA-tuning'). Our experiments demonstrated that LoRA tuning is a superior way of obtaining DP-synthetic data, which provides performance on the downstream task that is comparable or *even better* than directly DP-Training on real data. We showed that the standard NLP Prefix-LM loss is well suited for DP-finetuning. Private synthetic data can be used freely for all the purposes, such as feature engineering, hyperparameter tuning, debugging and monitoring, and sharing, but without any privacy-related concerns. We also showed that while Mauve is a good proxy metric for evaluating the quality of the synthetic data, simpler metrics like perplexity, when available, perform well.

# 7 ETHICS STATEMENT

We expect that our proposed method of generating DP synthetic data will facilitate safer data sharing and that the societal impact will be positive, since entities who own private data but do not necessarily

have the knowledge or resources to train predictive models can share private synthetic data with specialists for model creation, benefiting from their expertise without comprising the privacy of the users who contributed their data. The main limitation of our approach is that we only conducted experiments on English datasets, however we expect that methods should work on multilingual datasets as along as public multi-lingual data are available for LLM pre-training.

## 8 REPRODUCIBILITY STATEMENT

All of our experiments are based on open sourced frameworks and public datasets, refer to Appendices H and M. We further provided necessary details to reproduce our experiments in Appendices C, E, F, G and I.

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

# SUPPLEMENTARY MATERIAL

## A  OUR PRIVACY GUARANTEES

To provide all the information needed to understand our privacy guarantees, we follow the guidelines outlined in (Ponomareva et al., 2023).

1. **DP setting**. We provide central DP guarantee where the service provider is trusted to correctly implement the mechanism.

2. **Instantiating the DP Definition**

   (a) *Data accesses covered*: Our DP guarantees apply only to a single training run. We don't account for hyperparameter tuning in our guarantees.

   (b) *Final mechanism output*: We use DP-Training methods. Only model's predictions (e.g. synthetic data generated by the DP-Trained model) is released, however the mechanism's output is technically the full sequence of privatized gradients, and the guarantee also applies at this level. This also means that all the checkpoints are protected/can be released publicly.

   (c) *Unit of privacy*. We consider example-level DP, where each example is a long sequence of text, e.g. a full yelp review. Maximum length of such unit in tokens is 512, tokens are extracted by SentencePiece algorithm trained on c4. We consider full data protection (full text of a review).

   (d) **Adjacency definition for "neighboring" datasets"**: We use add-or-remove adjacency definition.

3. **Privacy accounting details**

   (a) *Type of accounting used*: RDP-based accounting.

   (b) *Accounting assumptions* : Poisson sampling was assumed for privacy amplification but shuffling was used in training)

   (c) *The formal DP statement*: We use various levels of $\epsilon$ values: 1,3,10. Our $\delta = \frac{1}{training\_data\_size}$

   (d) *Transparency and verifiability*: We are going to open source our code. We use open-sourced t5x framework.

## B  ADDITIONAL RELATED WORK

While in our paper we concentrate on generating synthetic text data, the task of generating synthetic tabular data has been explored extensively before.

**Synthetic Tabular data.**   For tabular data, early works on synthetic data concentrated on estimating the utility of the synthetic data as *a quality of the statistical answers* over the data (synthetic data for query release). Privacy-preserving aspect for such synthetic data was often achieved via DP methods (Blum et al., 2011; Hardt et al., 2010; Liu et al., 2021) More recently, works that instead evaluated how useful the tabular synthetic data is for some downstream ML model started gaining popularity (Tao et al., 2021). In general, the approaches for generating synthetic tabular data can be categorized into *Marginal-based* and *generative models-based*. Marginal-based models calculate private marginal distribution by taking marginal distribution over attributes and appropriately privatizing it with some DP-mechanism like Gaussian or Laplace. Then attributes for synthetic instances are sampled from these distributions. *Generative models* instead attempt to build a function that approximates the data distribution and sample from this function. GAN-based methods were a common choice for such generative models: e.g., DP-GAN (Xie et al., 2018), Convolutional GAN (Torfi et al., 2020) and PATE-GAN (Yoon et al., 2019). A recent benchmark (Tao et al., 2021) reported that for achieving best downstream ML model performance, marginal-based methods still outperform GAN-based approaches.

## C  LANGUAGE MODEL PRE-TRAINING.

**Model architecture.** For synthetic data generation, one can use either decoder only or encoder-decoder models. We used a decoder-only transformer model (Vaswani et al., 2017), which should be more parameter-efficient, with architecture similar to LaMDA (Thoppilan et al., 2022). The quality of the pre-trained model is of paramount importance for generating good fidelity synthetic data. We initially experimented with 1B model and found that a significant boost in downstream performance can be achieved by using a higher capacity model.

Therefore we use 8B model for final fine-tuning and prompt tuning. A smaller version (1B) model is used to tune hyperparameters like learning rate, clipping norm, batch etc. The hyperparameter values found using 1B model are used for the final 8B model. Statistics for 1B and 8B models are provided in the Table 5.

Table 5: Architecture of transformer models used in experiments.

| Parameters | Layers | Attn. heads | $d_{model}$ | $d_{ff}$ | $d_k, d_v$ |
|---|---|---|---|---|---|
| 1B | 8 | 32 | 2048 | 16384 | 128 |
| 8B | 16 | 64 | 4096 | 32768 | 128 |

**Pre-training data.** We used public dataset The Pile (Gao et al., 2020) as a basis for our pre-training data. However we run an extra post-processing step by deduplicating The Pile against all datasets used in downsteam tasks. This deduplication step is necessary to ensure that private data won't be accidentally learned during model pre-training, which otherwise would invalidate our DP guarantees.

To do deduplication we followed the recipe outlined in (Lee et al., 2022). Specifically, we tokenized and constructed a suffix array for each involved dataset (The Pile, IMDB, Yelp, AGNews). Then we used these suffix arrays to find common sequences of 50 of more tokens which appear in The Pile and any other dataset. Finally we cut all those common sequences from The Pile dataset. After cutting the sequences we de-tokenized dataset back to strings and used it for pre-training. It's important to note that we only deduplicate The Pile against other datasets, we did not run deduplication of The Pile against itself.

**Pre-training procedure.** We pre-trained our models using T5X codebase (Roberts et al., 2022), however we adopted few tricks which were used to train open sourced GPT-NeoX model (Black et al., 2022). Details are below.

We pre-trained a model for 380k steps using batch size of 1024 example with 1024 tokens sequence length, which result in training for approximately 400B tokens. We used same SentencePiece tokenizer which was used in original T5 model (Raffel et al., 2020). Cross entropy loss on next token prediction (teacher-forcing) was used as a training objective. Additionally we employed weight decay of 0.001 and an auxiliary z-loss $10^{-4} \log^2(Z)$ where $\log(Z)$ is softmax normalizer. Training was done with AdaFactor optimizer with learning rate schedule $\min(0.01, \frac{1}{\sqrt{N}})$ where $N$ is a step counter.

## D  ANALYSIS OF PRE-TRAINING DATA CONTAMINATION OF GPT-2 TRAINING DATA.

Multiple prior works on private synthetic data generation (Yue et al., 2022; Putta et al., 2023; Mattern et al., 2022) start with a pre-trained GPT-2 model and then finetune it with differential privacy on some downstream dataset. In this section we demonstrate that pre-training data for GPT-2 model includes examples from downstream tasks which invalidates the privacy guarantees of such prior work.

Let's assume we pre-train model $M$ on some public dataset $D_p$ and finetune with DP-SGD on sensitive dataset $D_s$. If $D_p \cap D_s = \emptyset$ then we can conclude that the process of obtaining our model $M$ is differentially private w.r.t. $D_s$. However this is no longer the case if intersection of $D_p$ and $D_s$ is non-empty.

GPT-2 was pre-trained on a WebText dataset (Radford et al., 2019). While the dataset is not released publicly, authors discuss in detail the process of how dataset was constructed by scraping the content

of links posted on Reddit website. Additionally, they released a list of top domains [2] from which the dataset was formed. Specifically, it could be seen that 183080 web-pages from IMDB and 36188 web-pages from Yelp websites were included GPT-2 pre-training data. This by itself, indicates that parts of IMDB and Yelp dataset were include in GPT-2 pre-training data.

We performed further analysis in the following way. We took a subset of OpenWebText dataset (Peterson et al., 2019) - a public re-implementation of WebText. Specifically we used `c4/webtextlike` from TFDS[3]. We computed an intersection of IMDB dataset and `c4/webtextlike` using the approach from (Lee et al., 2022). We found that `c4/webtextlike` contains 136 distinct examples from IMDB training and test set.

Finally we manually looked at a few of the examples to verify that they are indeed part of IMDB dataset and that they were likely used to construct WebText dataset. Here is a code snippet to obtain one of these examples:

```
import tensorflow_datasets as tfds
ds = tfds.load('imdb_reviews/plain_text:1.0.0',
               split='test',
               shuffle_files=False)
# get 52nd example from IMDB test set
print(next(iter(ds.skip(51)))['text'].numpy().decode('utf-8'))
# Output:
# In the eighties, Savage Steve Holland put out three movies ...

# This example is a second review from here:
# https://www.imdb.com/title/tt0091680/reviews
# The link to the IMDB web-page is mentioned
# in the following reddit post:
# https://www.reddit.com/r/movies/comments/77gna7/comment/dom5mqa
# The Reddit post was written in 2017, two years prior to
# creation of WebText dataset,
# suggesting that it would be included in the dataset.
```

This indicated that indeed at least some portion of the IMDB reviews would have been obtained during WebText dataset construction process and would have been used for GPT-2 pre-training. While it is impossible to say, without an access to the actual pretraining data, what percentage of the downstream tasks data was included for GPT-2, it highlight the importance of our deduplication process for obtaining rigorous privacy guarantees.

## E  DETAILS OF PROMPT-TUNING.

**Training instability with Adafactor optimizer and default prompt initialization.** (Lester et al., 2021) recommends Adafactor as a default optimizer. They also suggest to initialize prompt using pretrained embeddings of tokens from vocabulary. While we confirm that it works well for non-private prompt tuning of LLM, it appeared to be inadequate for DP-prompt tuning.

With this setup we observed significant training instabilities even in the best DP-runs after significant amount of clipping norm and learning rate tuning, see figure 2.

**Changing optimizer and prompt tuning initialization.**

To achieve good performance with prompt tuning we experimented with two things.. First of all, we tried various optimizers (including Adam and Momentum) instead of Adafactor. Additionally we experimented with random initialization of prompt tensor.

While changing optimizer didn't really improve the downstream performance, we found that Adam and Momentum optimizers lead to more stable training runs. Changing prompt tensor initalization to random uniform with small range was the key for improving prompt tuning performance with

---

[2]`https://github.com/openai/gpt-2/blob/master/domains.txt`
[3]`https://www.tensorflow.org/datasets/catalog/c4#c4webtextlike`

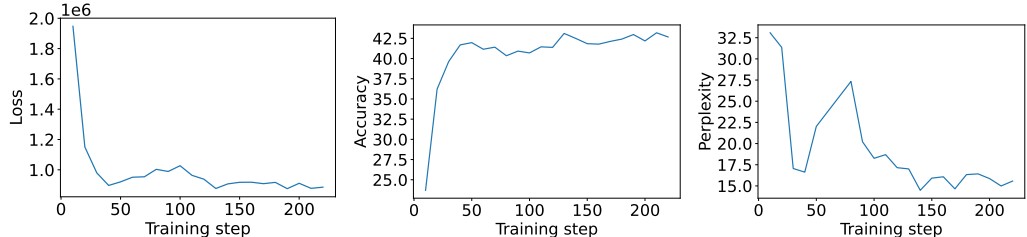

Figure 2: Training loss, training accuracy and validation perplexity for the best prompt tuning training run with Adafactor optimizer, $\epsilon = 1$.

DP. Table 6 demonstrates that random initialization results in a lift of up to 30% in downstream CNN performance, and up to 10% for BERT model. We hypothesize that proper initialization is very important to DP-Training and addition of noise makes it hard for the model to "recover" from bad initialization.

Table 6: Downstream performance of prompt tuning with different optimizers and prompt initialization. In all experiments prompt tuning was done for 10 epochs with 1024 batch size and privacy level $\epsilon = 1$. Column "Vocab Init" correspond to default initialization proposed in prompt tuning paper using embeddings of tokens from vocabulary. Column "Random Init" correspond to random uniform initialization in range $[-0.01, 0.01]$.

| Optimizer | BERT | | CNN | |
|---|---|---|---|---|
| | Vocab Init | Random Init | Vocab Init | Random Init |
| Adafactor | $72.1 \pm 2.1$ | $88.2 \pm 0.2$ | $55.8 \pm 0.2$ | $84.9 \pm 0.2$ |
| Adam | $74.6 \pm 1.2$ | $85.5 \pm 0.7$ | $50.2 \pm 0.1$ | $82.4 \pm 0.4$ |

We did some limited experiments with the scale of random uniform initialization, however for most of them we didn't run full synthetic data pipeline and only compared LLM perplexity and next token prediction accuracy. These experiments showed that random uniform from range $[-0.01, 0.01]$ was the best, while both decreasing and increasing the range led to worse metrics.

We also compared Adam optimizer with fixed learning rate, Adam optimizer with cosine learning rate decay and Momentum optimizer with cosine learning rate decay. As long as learning rate was tuned, all three performed similarly. Eventually we settled on Adam optimizer with fixed learning rate.

## F  DETAILS OF LORA

Let's say one of the layers in the network is represented as dense matrix multiplication operation $Wh$ where $W$ is a trainable weight and $h$ is an input to the layer (typically embedding or hidden state vector). LoRA (Hu et al., 2021) proposes to replace weight matrix with a sum $W + LR$, freeze $W$ and tune $L$ and $R$ matrices. In this notation $L$ and $R$ are low-rank matrices such that the result of their multiplication has the same shape as $W$. For example if $W$ is a matrix with the shape $n \times m$, then $L$ would be $n \times r$ matrix and $R$ would be $r \times m$ matrix, where $r$ is the rank of low-rank adapter.

Similar to (Hu et al., 2021), all layers where LoRA is not applied are considered frozen in our setup. We performed LoRA tuning using Adam optimizer with fixed learning rate. Unlike (Hu et al., 2021) we did not use weight decay because we observed no benefits of weight decay in differentially private LoRA training.

Typically, in a transformer model LoRA can be applied to attention layers and MLP layers. Hu et al. (2021) study LoRA only on attention layers. In this work, we tried to applied LoRA to attention layers only (Attention-LoRA), MLP layers only (MLP-LoRA) and both attention and MLP layers (Full-LoRA). Overall we found out that Full-LoRA is the best choice in most cases, however in some cases MLP-LoRA could be slightly better.

We also studied how rank of LoRA affects downstream performance. Similar to (Hu et al., 2021) we observed that initially increase of LoRA rank lead to increase of accuracy on downstream task, followed by eventual decrease of downstream accuracy with further increase of rank.

Our main results in table 1 are reported using Full-LoRA rank 8 on AGNews, rank 1 for Yelp datasets, and MLP-LoRA rank 32 on IMDB dataset.

Overall we can recommend to use Full-LoRA rank 8 as a good default value, however we can encourage tuning of LoRA parameters when resources allow it and the best possible performance is needed.

See also detailed ablation of LoRA parameters in Appendix J.5.

## G    DETAILS OF SYNTHETIC DATA SAMPLING.

For data generation/sampling part, LLMs have the following knobs:

1. *temperature*: this is a constant by which the logits are divided prior to softmax and subsequent sampling. Large temperature flatten the tokens distributions, making rarer tokens more likely to be selected; it also increases diversity of the data generated but can negatively affect the quality. Our experiments show that default temperature of 1 (so no modification of the tokens distribution) works the best.
2. *topk*: given a token distribution, topk determines what portion of the distribution to keep before sampling. Topk is similar to low temperature - e.g. setting topk to top 1000 tokens means that next token will be chosen from the most likely tokens. We keep this parameter unmodified (set to $\infty$).
3. *numdecodes*: is similar to the number of candidates in standard beam search. We use the value of 1 here, resulting in the sequence where each token is the most likely to be returned.

While we did experiment with various values of these hyperparameters (see appendix J.3), we find that default settings already provide enough of diversity of generated examples for our datasets. If further diversity needs to be enforced (e.g. for very large datasets), we recommend increasing the temperature or numdecode values.

## H    DETAILS OF DATASETS FOR DOWNSTREAM TASKS

We conducted experiments on IMDB reviews (Maas et al., 2011), Yelp reviews (Zhang et al., 2015a) and AGNews[4] datasets. On IMDB and Yelp we formulated downstream task as binary sentiment classification. On AGNews the downstream task was classification of titles of news articles into one of 4 topics (World, Sports, Business and Sci/Tech).

All of the considered datasets only provided training and test sets. To obtain validation set we split original training set into two chunks in deterministic way using TFDS split slicing API. We used first 90% of the original training set for training, and last 10% for validation.

Some of the dataset statistics for considered dataset and our train/test/validation splits is provided in table 7.

## I    ARCHITECTURE AND TRAINING OF DOWNSTREAM MODELS.

We used two types of models for all downstream experiments. First one is a BERT encoder with a dense layer on top of it, second one is a shallow CNN.

**BERT model.** For BERT-based classifier we used BERT-Base model (Devlin et al., 2018b) pretrained on English data[5] with standard BERT tokenization and preprocessing. We put a dense layer on top of pooled output of the BERT encoder to produce classification score. Output of dense layer was a single floating point number per input sequence which was converted to probability using sigmoid function. We trained entire model (including BERT encoder and dense layer) without freezing any layers.

**CNN model.** In addition to BERT, we used a shallow CNN model without any extra pre-training. Our model architecture followed the idea from (Johnson & Zhang, 2015) with the main difference

---

[4]https://www.tensorflow.org/datasets/catalog/ag_news_subset
[5]https://tfhub.dev/tensorflow/bert_en_uncased_L-12_H-768_A-12/4

Table 7: Datasets statistics.

|  | IMDB | Yelp | AGNews |
|---|---|---|---|
| Training examples | | | |
| Total | 22500 | 504000 | 108000 |
| Class 0 | 11256 | 252085 | 26915 |
| Class 1 | 11244 | 251915 | 26985 |
| Class 2 | - | - | 27081 |
| Class 3 | - | - | 27019 |
| Validation examples | | | |
| Total | 2500 | 56000 | 12000 |
| Class 0 | 1244 | 27915 | 3085 |
| Class 1 | 1256 | 28085 | 3015 |
| Class 2 | - | - | 2919 |
| Class 3 | - | - | 2981 |
| Test examples | | | |
| Total | 25000 | 38000 | 7600 |
| Class 0 | 12500 | 19000 | 1900 |
| Class 1 | 12500 | 19000 | 1900 |
| Class 2 | - | - | 1900 |
| Class 3 | - | - | 1900 |

that we didn't pre-train embeddings beforehand. To be more specific, our model works as follows. We convert input sequence to lowercase and tokenize it by splitting it on whitespaces and punctuation signs. Then we embed it into 384 dimensional embedding, using vocabulary from most common 30k words from the dataset. Embedding was randomly initialized and trained as a part of downstream training task. Output of embedding layer was passed through 1D convolution with kernel size 3, 256 output filters and RELU activation. This followed by global max pooling along the sequence length. Then we have a fully connected layer with 256 outputs and RELU activation, followed by final logits layer with sigmoid activation.

**Training of downstream model**. Both BERT and CNN were trained in a similar way. For non-private training we used Adam optimizer and DP-Adam (as implemented in the Keras DPModel library) for private training. Model was regularized with weight decay and we did early stopping (by non-increasing validation accuracy). We choose best hyperparameters by running a sweep over learning rates $\{10^{-7}, \ldots, 10^{-1}\}$ and weight decays $\{5 \times 10^{-1}, \ldots, 5 \times 10^{-6}, 0\}$. Additionally for DP-training baseline on real data we did a sweep of clipping norm over $\{10^{-2}, \ldots, 10^2\}$.

## J  ABLATIONS

We study the influence of various factors on quality of generated synthetic data. We observed that better pre-trained model (in terms of model size and choice of pre-training dataset) leads to better synthetic data in differentially-private case, see details in Appendix J.1.

All standard techniques which are used to improve utility of DP-training (Ponomareva et al., 2023) apply to both full fine-tuning and parameter-efficient tuning of LLMs. Specifically, it's usually better to train longer and with larger batch size (Ponomareva et al., 2023). Also it's important to do a hyperparameter sweep over learning rate and clipping norm, see appendix J.2. We also experimented with various parameters of temperature sampling for data synthesis. We found that sampling with $T = 1$, without truncating sampling distribution (i.e. TopK $= \infty$) and without performing any filtering usually well and is a reasonable default setting, refer to Appendix J.3.

We also looked into variations of loss formulation for LLM training (Appendix J.4). As explained in Section 4.1 we found that Prefix-LM (Raffel et al., 2020) formulation usually leads to better performance in DP setting, compared to Full LM. We observed that normalizing LLM loss by number of non-padding tokens, as suggested in (Ponomareva et al., 2022), makes it easier to tune LLM hyperparameters, especially clipping norm for DP. It also makes it easier for LLM to learn to produce outputs of various length.

## J.1 ABLATION: PRE-TRAINING DATASET AND MODEL SIZE

We looked into influence of LLM pre-trainind dataset and model size on performance on downstream task. As shown in table 8 larger size of LLM translated into better downstream performance for both BERT and CNN models.

Table 8: Influence of model size on downstream performance. All models are finetuned or prompt tuned for 10 epochs with 1024 batch size. Prompt tuning is done with loss normalization, fine-tuning is without. DP level is $\epsilon = 1$.

| Model size | Prompt tune | | Fine tune | |
|---|---|---|---|---|
| | BERT | CNN | BERT | CNN |
| 1B | $83.2 \pm 1.1$ | $76.9 \pm 0.3$ | $74.7 \pm 2.4$ | $61.1 \pm 0.6$ |
| 8B | $86.2 \pm 0.4$ | $83.2 \pm 0.4$ | $85.5 \pm 0.2$ | $81.7 \pm 0.3$ |

Additionally, we did some limited study comparing pre-training on C4 and The Pile datasets. Initially we expected that C4 (which is essentially a web crawl) better matches text distribution in Yelp and IMDB datasets compared to The Pile (which was intentionally composed to be more diverse). However our experiments showed similar performance on both datasets, so eventually we settled on The Pile, which is easier to download and use.

## J.2 ABLATION: HYPERPARMETER TUNING FOR DP-TRAINING OF LLM

**Batch size and number of training steps.** For our prompt tuning run on IMDB with $\epsilon = 1$ we did a thorough sweep of various batch sizes and number of training steps of LLM prompt tuning. Results are summarized in table 9 and confirm the general observation that longer training with larger batch is typically better when trained with DP.

Table 9: This table show how downstream performance depends on batch size and number of training steps used for LLM prompt tuning. Experiments were done on IMDB task, synthetic data were generated at privacy level $\epsilon = 1$. Noise multiplier for each training run was computed to satisfy privacy budget given chosen number of steps and batch size. Batch size 1024 with 220 steps correspond to 10 epochs of LLM training on IMDB dataset.

| Batch size | BERT | | | CNN | | |
|---|---|---|---|---|---|---|
| | 220 steps | 440 steps | 880 steps | 220 steps | 440 steps | 880 steps |
| 1024 | $85.5 \pm 0.7$ | $85.1 \pm 0.6$ | $87.8 \pm 0.2$ | $82.4 \pm 0.4$ | $83.2 \pm 0.4$ | $84.3 \pm 0.1$ |
| 2048 | $85.7 \pm 0.2$ | $86.4 \pm 0.9$ | $87.9 \pm 0.1$ | $83.8 \pm 0.2$ | $83.2 \pm 0.2$ | $85.0 \pm 0.2$ |
| 4096 | $86.0 \pm 0.6$ | $86.6 \pm 0.2$ | $88.1 \pm 0.4$ | $82.8 \pm 0.4$ | $83.8 \pm 0.2$ | $85.4 \pm 0.1$ |

**Learning rate.** In our experiments we found that downstream performance was quite sensitive to learning rate of LLM, see table 10.

Table 10: Sensitivity of downstream task performance to learning rate used for LLM tuning.

(a) Finetuning

| Learning rate | BERT | CNN |
|---|---|---|
| 1e-2 | Diverge | |
| 3e-3 (optimal) | $85.5 \pm 0.2$ | $81.7 \pm 0.3$ |
| 1e-3 | $69.1 \pm 3.0$ | $53.1 \pm 0.8$ |

(b) Prompt tuning

| Learning rate | BERT | CNN |
|---|---|---|
| 3e-3 | $83.4 \pm 0.5$ | $80.0 \pm 0.4$ |
| 1e-3 (optimal) | $86.2 \pm 0.4$ | $83.2 \pm 0.4$ |
| 3e-4 | $84.7 \pm 0.9$ | $80.1 \pm 0.6$ |

## J.3 ABLATION: SAMPLING PARAMETERS.

After training a generative model, we still need to use that model to create a differentially private dataset. Large language models generate sequences one tokens at a time in an autoregressive manner; given previous tokens the model's final layer emits a probability distribution over all possible next tokens. Instead of sampling directly from this probability distribution, it is common to modify it in three ways:

1. **Temperature**: Shaping the token distribution using temperature $t$ so that the final softmax over the final logits $u_i$ gives the next token probably as

$$P(z_i|z_{<i}) = \frac{exp(u_i/t)}{\sum_i exp(u_i/t)} \tag{2}$$

2. **Top K**: Truncating the token distribution so that only the $k$ most likely tokens are sampled from. All other tokens are given zero probability.

3. **Num Decodes**: Repeating the full sampling process (i.e. decoding) N times and then out of the N candidates return only the sample with the highest likelihood.

To analyze the effect of these parameters on dataset quality we performed a sweep over these parameters and computed the downstream performance of models as discussed in section I. Results are shown in tables 11 and 12.

Results from our ablation study on sampling parameters show that while small gains can be gained from tuning these parameters, such gains are modest compared to using the default parameters of $t = 1.0$, top-k $= \infty$, and num decodes $= 1$. We note that slightly higher temperatures appear to help in both cases (1.4 for IMDB and 1.2 for Yelp) and in both cases should be paired with either tighter top-k or an increase in decodes. However, using additional decodes are computationally costly and likely not worth the additional cost. Using a temperature less than 1.0 never seems to help.

### J.4    ABLATION: LOSS OF LLM.

LLMs are commonly trained/fine-tuned with next-token prediction teacher forcing. The loss for each token in this setup is a cross-entropy loss for each token. For an instance that is a collection of tokens (e.g. a full yelp review), the loss is therefore is a sum over per-token losses. It is common to normalize this loss by the number of non padding tokens, which roughly translates into the normalization by the batch size and the average number of tokens for instances in the batch. We recommend to follow this normalization scheme because it makes it much easier to find the appropriate clipping norm for DP-Training. When the loss is not normalized, the appropriate clipping norm can be in the thousands. With normalized loss, a standard clipping norm of 1 or 3 usually works out of the box. For example, for Yelp dataset, without the loss normalization the clipping norm was found to be approx 2000, with accuracy of the fine-tuning of approx 0.29. With loss normalization, the clipping norm of 1 resulted in performance of 0.43.

Note that if it's feasible to perform full hyperparameter sweep of clipping norm and learning rate, then benefits of loss normalization diminishes. Nevertherless even in this case loss normalization can provide small advantage, see Table 13.

### J.5    ABLATION: LORA PARAMETERS.

We conducted detailed sweep of LoRA parameters (rank and in which layers to introduce LoRA) on IMDB and AGNews, see Tables 14 and 15. As could be seen from these tables, Full-LoRA performs better than Attention-LoRA and MLP-LoRA in most cases on IMDB dataset. However MLP-LoRA seem to be better choice overall on AGNews. If practitioner has to pick parameters to introduce LoRA beforehand without tuning, then we would recommend Full-LoRA as a reasonable default.

In terms of rank, best performance is typically achieved for ranks in range $[8, 32]$. From our experiments, it does not make sense to increase rank above 32 because it result in little to no performance gains, however it is more expensive because requires tuning of more parameters.

### K    EVALUATING SYNTHETIC DATA QUALITY: MAUVE ROBUSTNESS.

Mauve has multiple parameters that control its behavior. The most influential such parameters include: the degree of dimensionality reduction (PCA explained variance), the number of clusters to use, the number of samples to use, and the model used to initially embed the samples. Pillutla et al. (2021) came to the conclusion that while these affected performance none of these parameters mattered enough to worry about needing to tune for a specific application. They recommended a default setting of buckets $= 500$ and explained variance $= 0.9$, We performed our own ablation studies on these

Table 11: Comparison across sampling parameters for best performing prompt-tuned epsilon=1 model for the **IMBD** task. The parameters Temp = 1.0, TopK = ∞, Decodes = 1 corresponds to the default used in this paper.

| Temp | TopK | Decodes | BERT Accuracy | CNN Accuracy |
|---|---|---|---|---|
| 0.6 | ∞ | 1 | $80.7 \pm 1.4$ | $78.4 \pm 1.2$ |
|  |  | 2 | $78.7 \pm 1.8$ | $76.9 \pm 1.4$ |
|  |  | 4 | $79.5 \pm 3.0$ | $76.3 \pm 1.6$ |
|  | 100 | 1 | $81.1 \pm 0.7$ | $77.9 \pm 1.5$ |
|  |  | 2 | $78.1 \pm 1.1$ | $77.8 \pm 0.5$ |
|  |  | 4 | $79.9 \pm 1.0$ | $75.9 \pm 0.7$ |
|  | 1000 | 1 | $79.9 \pm 0.7$ | $77.1 \pm 1.4$ |
|  |  | 2 | $78.7 \pm 1.4$ | $75.4 \pm 2.4$ |
|  |  | 4 | $78.1 \pm 2.3$ | $77.8 \pm 1.3$ |
| 0.8 | ∞ | 1 | $84.7 \pm 0.5$ | $82.2 \pm 0.9$ |
|  |  | 2 | $83.5 \pm 1.4$ | $82.8 \pm 0.4$ |
|  |  | 4 | $82.8 \pm 0.4$ | $81.9 \pm 1.0$ |
|  | 100 | 1 | $84.9 \pm 0.6$ | $82.2 \pm 0.6$ |
|  |  | 2 | $85.4 \pm 1.1$ | $81.6 \pm 0.7$ |
|  |  | 4 | $82.1 \pm 1.0$ | $82.9 \pm 1.0$ |
|  | 1000 | 1 | $84.1 \pm 1.0$ | $82.2 \pm 0.2$ |
|  |  | 2 | $82.9 \pm 1.0$ | $82.1 \pm 0.9$ |
|  |  | 4 | $82.1 \pm 1.1$ | $83.1 \pm 0.3$ |
| 1.0 | ∞ | 1 | $85.3 \pm 0.4$ | $84.2 \pm 0.6$ |
|  |  | 2 | $86.1 \pm 0.2$ | $83.2 \pm 0.1$ |
|  |  | 4 | $85.4 \pm 0.4$ | $83.5 \pm 1.7$ |
|  | 100 | 1 | $84.5 \pm 0.7$ | $84.9 \pm 0.4$ |
|  |  | 2 | $86.8 \pm 0.4$ | $84.5 \pm 0.3$ |
|  |  | 4 | $84.4 \pm 0.9$ | $84.7 \pm 0.3$ |
|  | 1000 | 1 | $86.3 \pm 0.4$ | $83.7 \pm 1.0$ |
|  |  | 2 | $86.5 \pm 1.7$ | $84.3 \pm 0.3$ |
|  |  | 4 | $85.6 \pm 0.4$ | $84.2 \pm 0.3$ |
| 1.2 | ∞ | 1 | $87.1 \pm 0.7$ | $77.4 \pm 1.2$ |
|  |  | 2 | $86.0 \pm 1.7$ | $78.5 \pm 2.9$ |
|  |  | 4 | $85.6 \pm 0.9$ | $79.9 \pm 1.5$ |
|  | 100 | 1 | $85.8 \pm 1.3$ | $84.4 \pm 0.1$ |
|  |  | 2 | $87.2 \pm 0.4$ | $84.8 \pm 0.6$ |
|  |  | 4 | $86.5 \pm 1.0$ | $85.2 \pm 0.2$ |
|  | 1000 | 1 | $86.7 \pm 0.2$ | $81.9 \pm 0.2$ |
|  |  | 2 | $86.9 \pm 0.6$ | $81.9 \pm 0.7$ |
|  |  | 4 | $86.4 \pm 0.3$ | $82.3 \pm 0.5$ |
| 1.4 | ∞ | 1 | $84.1 \pm 1.6$ | $60.0 \pm 3.5$ |
|  |  | 2 | $85.5 \pm 1.2$ | $66.1 \pm 1.1$ |
|  |  | 4 | $81.6 \pm 2.0$ | $60.6 \pm 2.9$ |
|  | 100 | 1 | $86.5 \pm 0.6$ | $83.3 \pm 0.2$ |
|  |  | 2 | $86.9 \pm 0.3$ | $84.3 \pm 0.2$ |
|  |  | 4 | $85.8 \pm 0.9$ | $84.4 \pm 0.2$ |
|  | 1000 | 1 | $85.3 \pm 1.2$ | $77.2 \pm 2.2$ |
|  |  | 2 | $86.4 \pm 0.2$ | $78.3 \pm 0.8$ |
|  |  | 4 | $86.6 \pm 0.4$ | $79.2 \pm 0.4$ |

parameters (see Figure 3). As noted in the paper, while we found MAUVE to be robust to most of these parameters, the model mattered a great deal.

## L    EVALUATING SYNTHETIC DATA QUALITY: PROXY METRIC EXPERIMENTS.

Proxy metrics are a less expensive method for estimating synthetic data quality. They are particularly useful for helping tune the large number of hyperparameters needed to train and sample from the generator model. This includes tuning the model type (model architecture, pre training process, fine-tuning, prompt-tuning, etc..), the hyper parameters of model training (e.g. epsilon, clipping norm, learning rate, batch size, epochs, etc..), and finally the hyperparameters of the sampling procedure

Table 12: Comparison across sampling parameters for best performing prompt-tuned epsilon=1 model for the **Yelp** task. The parameters Temp = 1.0, Top-K = $\infty$, Decodes = 1 corresponds to the default used in this paper.

| Temp | Top K | Decodes | BERT Accuracy | CNN Accuracy |
|------|-------|---------|---------------|--------------|
| 0.8 | $\infty$ | 1 | $92.6 \pm 0.6$ | $87.9 \pm 1.1$ |
| | | 2 | $89.9 \pm 1.1$ | $85.0 \pm 1.3$ |
| | | 4 | $88.3 \pm 0.1$ | $81.7 \pm 0.9$ |
| | 100 | 1 | $91.7 \pm 0.6$ | $86.6 \pm 1.6$ |
| | | 2 | $89.9 \pm 0.6$ | $84.2 \pm 1.5$ |
| | | 4 | $87.5 \pm 1.6$ | $80.5 \pm 0.4$ |
| | 1000 | 1 | $92.5 \pm 0.3$ | $87.9 \pm 0.6$ |
| | | 2 | $90.4 \pm 1.4$ | $85.4 \pm 1.7$ |
| | | 4 | $86.9 \pm 1.1$ | $82.5 \pm 1.9$ |
| 1 | $\infty$ | 1 | $93.4 \pm 0.7$ | $90.9 \pm 0.5$ |
| | | 2 | $93.9 \pm 0.5$ | $90.5 \pm 0.4$ |
| | | 4 | $93.7 \pm 0.6$ | $90.5 \pm 0.3$ |
| | 100 | 1 | $93.2 \pm 0.5$ | $90.0 \pm 0.5$ |
| | | 2 | $93.3 \pm 0.2$ | $89.4 \pm 0.6$ |
| | | 4 | $92.4 \pm 0.2$ | $88.3 \pm 0.3$ |
| | 1000 | 1 | $93.9 \pm 0.2$ | $90.8 \pm 0.2$ |
| | | 2 | $94.0 \pm 0.2$ | $91.1 \pm 0.1$ |
| | | 4 | $93.8 \pm 0.2$ | $89.7 \pm 0.6$ |
| 1.2 | $\infty$ | 1 | $93.6 \pm 0.3$ | $90.1 \pm 0.2$ |
| | | 2 | $93.5 \pm 0.2$ | $90.3 \pm 0.0$ |
| | | 4 | $94.0 \pm 0.1$ | $90.6 \pm 0.1$ |
| | 100 | 1 | $93.7 \pm 0.2$ | $91.2 \pm 0.1$ |
| | | 2 | $93.7 \pm 0.1$ | $91.3 \pm 0.3$ |
| | | 4 | $93.8 \pm 0.2$ | $91.3 \pm 0.2$ |
| | 1000 | 1 | $94.1 \pm 0.1$ | $90.7 \pm 0.2$ |
| | | 2 | $94.0 \pm 0.3$ | $90.9 \pm 0.1$ |
| | | 4 | $94.2 \pm 0.1$ | $91.0 \pm 0.1$ |
| 1.4 | $\infty$ | 1 | $93.2 \pm 0.2$ | $86.2 \pm 1.7$ |
| | | 2 | $93.1 \pm 0.4$ | $87.2 \pm 0.8$ |
| | | 4 | $92.9 \pm 0.8$ | $86.7 \pm 0.6$ |
| | 100 | 1 | $93.4 \pm 0.4$ | $90.8 \pm 0.0$ |
| | | 2 | $93.5 \pm 0.4$ | $90.9 \pm 0.0$ |
| | | 4 | $93.5 \pm 0.0$ | $91.0 \pm 0.0$ |
| | 1000 | 1 | $93.4 \pm 0.1$ | $88.9 \pm 0.1$ |
| | | 2 | $93.5 \pm 0.0$ | $89.2 \pm 0.1$ |
| | | 4 | $93.4 \pm 0.5$ | $89.2 \pm 0.3$ |

Table 13: Effect of loss normalization with sufficient hyperparameter tuning.

| | BERT | | CNN | |
|---|------|---|-----|---|
| | Loss Norm | No Loss Norm | Loss Norm | No Loss Norm |
| IMDB prompt tuning, $\epsilon = 1$ | $86.4 \pm 0.9$ | $86.0 \pm 0.7$ | $83.2 \pm 0.2$ | $83.0 \pm 0.5$ |

needed to create the final dataset (e.g. temperature, top-k, and decodes). We examined how well various metrics correlate with downstream performance of a final classifier trained on the synthetic data (Figure 4). Since we are interested in selecting the best performing parameters, we want a metric that is most likely to select a high quality model and thus should care primarily about the rank correlation.

## L.1 SYNTHETIC DATA TEXT LENGTH COMPARISON

While the distribution of text lengths isn't the most correlated to downstream performance, it still has strong predictive power, and has the advantage of being easy to visualize and understand. We plot the distribution of lengths across all final datasets (across architecture and epsilon) for the IMDB and YELP datasets (figures 5 and 6). We would point out that all synthetic data is truncated to 512 tokens because this was sequence length of the trained LLM. At the same time, some of the real data is longer (17% of imdb and 7% of yelp) .

Table 14: Performance of downstream classifier depending on LoRA parameters, when LLM is trained on IMDB dataset with $\epsilon = 1$.

| | LoRA | Rank | | | | | | | |
|---|---|---|---|---|---|---|---|---|---|
| | | 1 | 2 | 4 | 8 | 16 | 32 | 48 | 64 |
| BERT | Full | $85.4 \pm 0.2$ | $85.9 \pm 0.3$ | $89.7 \pm 0.1$ | $90.0 \pm 0.3$ | $90.0 \pm 0.2$ | $89.7 \pm 0.2$ | $89.9 \pm 0.3$ | $90.2 \pm 0.2$ |
| BERT | MLP | $84.4 \pm 0.5$ | $85.9 \pm 0.3$ | $88.2 \pm 0.4$ | $86.1 \pm 0.1$ | $88.2 \pm 0.5$ | $87.9 \pm 0.2$ | $86.1 \pm 0.2$ | $88.1 \pm 0.7$ |
| BERT | Attn | $82.5 \pm 0.4$ | $88.8 \pm 0.1$ | $87.7 \pm 0.3$ | $89.7 \pm 0.3$ | $90.1 \pm 0.1$ | $89.0 \pm 0.3$ | $89.8 \pm 0.3$ | $89.6 \pm 0.1$ |
| CNN | Full | $75.5 \pm 0.8$ | $84.1 \pm 0.7$ | $85.5 \pm 0.2$ | $87.6 \pm 0.4$ | $87.4 \pm 0.3$ | $87.2 \pm 0.2$ | $87.3 \pm 0.2$ | $87.5 \pm 0.3$ |
| CNN | MLP | $71.9 \pm 0.6$ | $81.7 \pm 0.4$ | $86.8 \pm 0.1$ | $81.6 \pm 0.9$ | $85.7 \pm 0.2$ | $85.5 \pm 0.2$ | $83.6 \pm 0.3$ | $85.1 \pm 0.3$ |
| CNN | Attn | $72.2 \pm 1.9$ | $86.9 \pm 0.3$ | $84.0 \pm 0.1$ | $87.4 \pm 0.2$ | $87.3 \pm 0.1$ | $85.7 \pm 0.1$ | $87.3 \pm 0.1$ | $86.8 \pm 0.4$ |

Table 15: Performance of downstream classifier depending on LoRA parameters, when LLM is trained on AGNews dataset with $\epsilon = 1$.

| | LoRA | Rank | | | | | | | |
|---|---|---|---|---|---|---|---|---|---|
| | | 1 | 2 | 4 | 8 | 16 | 32 | 48 | 64 |
| BERT | Full | $88.1 \pm 0.1$ | $88.0 \pm 0.1$ | $88.2 \pm 0.3$ | $88.5 \pm 0.1$ | $87.9 \pm 0.1$ | $88.3 \pm 0.2$ | $88.8 \pm 0.1$ | $88.9 \pm 0.1$ |
| BERT | MLP | $87.9 \pm 0.1$ | $88.0 \pm 0.2$ | $88.2 \pm 0.0$ | $88.4 \pm 0.1$ | $88.8 \pm 0.3$ | $89.4 \pm 0.1$ | $88.3 \pm 0.2$ | $88.3 \pm 0.1$ |
| BERT | Attn | $87.7 \pm 0.2$ | $88.4 \pm 0.1$ | $88.1 \pm 0.2$ | $88.0 \pm 0.3$ | $88.2 \pm 0.1$ | $88.7 \pm 0.1$ | - | - |
| CNN | Full | $85.3 \pm 0.1$ | $85.0 \pm 0.1$ | $85.2 \pm 0.1$ | $85.4 \pm 0.1$ | $84.6 \pm 0.2$ | $85.3 \pm 0.1$ | $85.6 \pm 0.1$ | $85.7 \pm 0.2$ |
| CNN | MLP | $84.9 \pm 0.2$ | $85.0 \pm 0.1$ | $85.2 \pm 0.1$ | $85.7 \pm 0.1$ | $85.6 \pm 0.2$ | $85.8 \pm 0.0$ | $85.2 \pm 0.1$ | $85.2 \pm 0.1$ |
| CNN | Attn | $85.2 \pm 0.1$ | $85.3 \pm 0.1$ | $84.6 \pm 0.1$ | $85.0 \pm 0.1$ | $85.1 \pm 0.1$ | $85.4 \pm 0.1$ | - | - |

## L.2 SYNTHETIC DATA BIGRAM FREQUENCY COMPARISON

Comparing the distribution if bigrams across datasets can give additional insight into how the synthetic data differs between finetuning approach and epsilon. As one might expect, less noise leads to a smaller distributional gap (figure 7). Full finetuning has the largest gap, followed by prompt tuning, and LoRA being the most aligned with the original distribution.

## M  IMPLEMENTATION DETAILS AND ESTIMATES OF THE REQUIRED COMPUTE

**LLM training.** We run all our experiments using T5X codebase and used implementation of transformer layers from Flaxformer library. For differentially private training of transformers we used private_text_transformers repository.

We pre-trained 1B model on TPUv3 with 128 cores and 8B model was pretrained on TPUv3 with 1024 cores, both runs took around 8 days. Both finetuning and prompt-tuning of 8B models was done on TPUv3 with 128 cores. Diferentially private finetuning of 8B model required between 20 hours (for shortest run on IMDB) and up to 80 hours for some of the longer runs. Prompt tuning required 1.5 hour for short run and up to 20 hours for longest runs. LoRA-tuning was about 2x slower compared to prompt tuning with 3 hours training for shortest run and up to 2 days for longest runs.

In most experiments we used a total batch size of 1024 examples per optimizer step, however we did run a few ablations with larger batch as well. In order to fit entire batch into memory we used a technique called gradient accumulation. This technique splits entire batch into smaller chunks, sequentially computes gradients over each chunk and then aggregates them together to obtain final gradient for entire batch. We used chunks of 32 examples for full finetuning and LoRA, and chunks of size up to 256 for prompt tuning. Even with difference in chunk sizes, we observed that LoRA is only 2x slower compared to prompt tuning when using the same total effective batch and same number of training steps.

**Downstream model.** Downstream classifier was implemented in Tensorflow using Keras library for training and TFDS to load datasets.

Each downstream model was run on TPUv2 with 8 cores. To obtain each downstream accuracy number we run a sweep of around 28 different hyperparameter settings. Entire sweep took around 4 hours for CNN model and up to 80 hours combined for BERT model and synthetic dataset of 0.5M examples. Each sweep was repeated 3 times to compute error bars.

## N  EXAMPLES OF GENERATED AND REAL DATA

Table 16 shows examples of generated synthetic data.

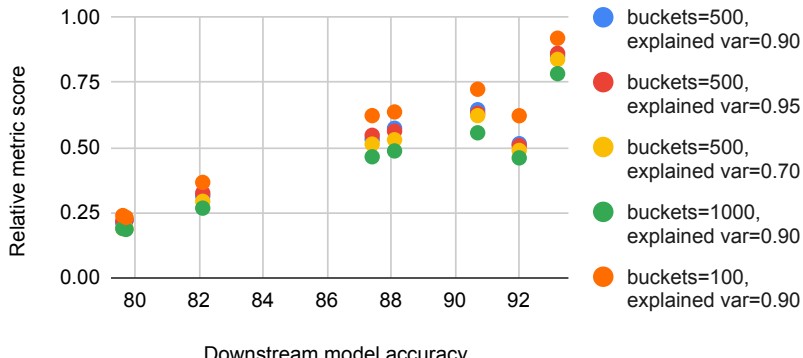

Figure 3: Example of varying MAUVE parameters on estimating IMDB downstream performance on datasets differing in training epsilons from Table 1. Results are shown for the Sentence-T5-8B model.

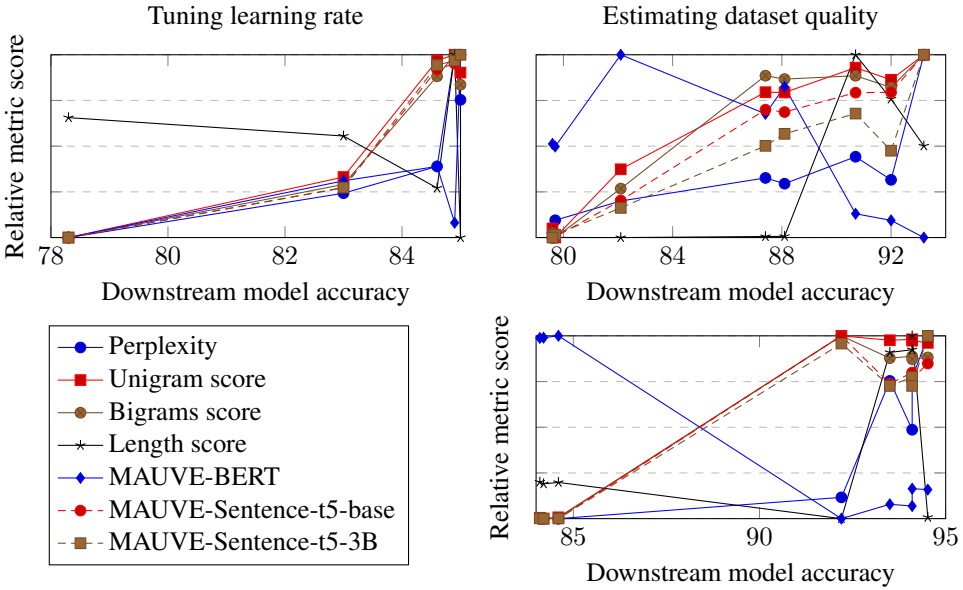

Figure 4: Proxy metrics for estimating dataset quality. Each point represents a metric's estimate of a synthesized dataset plotted against its true downstream classifier performance. X-axis shows the metric values re-scaled. All metrics are only useful to compare datasets, and thus their absoluteabsolute value is uninformative. Top Left: Different learning rates of an IMDB prompt-tuning model. Top Right: Estimating IMDB dataset quality for results in Table 1. Bottom Right: Estimating Yelp dataset quality for results in Table 1.

Table 16: Examples of real and synthetic datasets.

| Eps | Dataset | Class | Example |
|---|---|---|---|
| ∞ (real) | Yelp | Negative | Mediocre burgers - if you are in the area and want a fast food burger, Fatburger is a better bet than Wendy's. But it is nothing to go out of your way for. |
| | | Negative | Not at all impressed...our server was not very happy to be there...food was very sub-par and it was way to crowded. Not the good kind I crowded where you feel like \""wow this is great it must be busy because the food is so great..\"" But the type of crowded where you feel a fight may break out.  Also, if the chocolate fountain is their golden gem...why is it ok for people to dip the strawberry,lick off the chocolate and re dip it right then and there...absolutely disgusting! I Waited almost 10 minutes to dip...saw that and was immediately turned away...just saying...Never again! |
| | | Positive | I know, I know a review for Subway, come on. But I have to say that the service at this subway is top notch. The staff is very friendly and always goes out of there way to say hello. It is highly appreciated. The food is pretty standard Subway fare. Not too much to say there, but I will be be back again on my lunch breaks. |

| | | | |
|---|---|---|---|
| ∞ (synthetic) | Yelp | Negative | I waited for a table to be seated but it was not until 45 minutes into my 45 minute lunch. The service and attention to detail is outstanding. I got served by one gentleman which was fine until another needed help bringing an food order. He was not willing to ring me back until the staff had technically completed more than just a couple steps in the process. This was one of the most memorable bad encounters I have had with service anywhere. It was so beyond poor that I couldn't even settle on a 1 star review for the sole purpose of the bad service.This is managements reptile effect. I may even review this restaurant on the yelp website as part of the bad service. ? r ? n ? r ? nWhat i do remember about the food was that I got the (medium well)lamb chops which my family enjoyed. I believe this was the only 4.5 star review; all others are 5 stars. ? r ? n ? r ? nThank you Taste of Thai for being so exceptional. I will definitely eat here again.\" 2083 S. Colorado Ave. ? r ? nLansing, MI 492019 ? r ? nEmail: revgiebun@yahoo.com ? r ? n ? r ? n\"[eod] [pile] ; RUN: llc,?,%s -mtriple=sparclite-serializer-macho -o /dev/null ; The powerpc 64 bit ABI hard float ABI contains 32-bit values types ; (e.g. 6 (struct half*) or 7 (struct bfin64_2float3216*)). The Cortex-A8 ; processor doesn't contain facilities to load bulky 32-bit types (it only ; has equivalents for 8-bit and 16-bit types and it joined processing ; these groups into the same registers) ; RUN: ld.ll,?,%s,?,%p/../Inputs/structures.s ; RUN: llvm-objdump -h -j ../Inputs/structures.exe.macho.x86_64 -b binary |
| | | Negative | Restaurant is not worth the drive other places near Ames and Des Moines have better options. Service was weird and owner was a jerk for not doing anything with it. Saw some actual fish caught and skinned by people. They would take their picture just put it,on the wall. Submitted by: TomRindal A Restaurant Review of Guadalabar [yelp],?,?,?,?,? ! Food could be a lot tastier if we paid more for the whole dining experience. They only charged me for one third of my dish and always put a kick in the case. However, the fish and ric e?! Huh? Fish A..a..er... jiabitangellois so so bad..and it's served on top of the rice which kills all of the yuminess. There are still so many great places to eat in this town, just need a little more energy to do it MrTripA4NO beer!!!! Good place to bring children... a dinky, cr appy fake Spongebob movie saturday morning in a 10 year old's restaurant.... been to Des Moines a few times and tried to find something tastier then Guadalabar, but at least they had smoothies.. and know what you are getting before you overpay.. A great nd cheap breakfast! $2 sandw ich for \"bonus\"... A calligraphy class flyer!! wow.. our waiter was the dishwasher; the restaurant was dirty & smelled like wet concrete. Definitely would not recommend.... this place has no charm... Low Yelp rating = the customer service and manager are living up the secret. May be its just another Ogden this time next year. Next time demote after 3rd year.,? ! ?,?,? cheesy ? ! ? ! Had three dinners at Guadalabar. Last one was last thurs, Wow! The pig in a blanket is outstanding! If lamb n onions don't click your menu versus noodle risotto is a clear,must. My partner got the cheese appetizer. Decent! The fish sans name was tasty, and definitely |
| | | Positive | Super cool food and interesting menu. Anything or everything can be made vegan and they are more than willing to cater to the vegan individual or families. I went in with a group of sisters who aren't into tofu and cepes on a regular basis. They were thrilled! I ordered the Mixed Marinara Veratofu which its made with squash, gnocchi, mushrooms, onions and is all natural. The Greens are also amazing, with quinoa and not so many red charcters like burgers. Look forward to this place anytime. ? n The best part is that they are great to the environment ? n Great service (despite the fact it took a long, long time for the food to arrive), and low prices. Great food and it is away from downtown SF ? n We are fans! ? n ? n ? n Florence 2232 S Broadway #102 ? n Studio City, CA 91306 (562) 285-9947 http://theamazingvegetemenu.com ? n ? n ? n DTLA #100 ? n Serving: Late Night Until 2 AM ? |
| 1 | Yelp | Negative | Best bahn in Belgium 50 years ago and 50 years later. First, the breakfast is awful and thin. Reviews on yelp keep complaining. Most certainly were never sold in Poland. The taste is not bread at all, but as crumbly and flat pancakes filled with sugar. |
| | | Negative | If you are planning a party night out, don't go here. They back stab you, yell at you to leave the store, and are rude and rude. I told my mom that if we went to Frisch's for party supplies she would stay away a good 10 miles away. Thank goodness for Walmart. |
| | | Positive | I am actually surprised they weren't busy like Wendi tweeted me. What a beautiful restaurant and atmosphere! Please take my advice and make sure you stroll around golf course for some free and attractive restaurants, shops and more. |
| | | Positive | OMG, WOW, WHAT IS THIS?? Go for the Thai Iced Tea. My usual order is iced capp and has tea but this is better. Reliable, inexpensive, and delicious! I have never had this before and this was a revelation! |
| 3 | Yelp | Negative | Good choice, bad service. I live in Aliso Viejo and the service at Windrider Station has always been great but our experience this evening was profoundly disappointing. The services mood, warmth, and grace is what set Windrider Station services apart from the rest, so I was surprised to see it be rendered to an adept officer. |
| | | Negative | Been to the place a few times for Detroit wings. The taco meat was uneatable.. .carelessness in processing.. seriously.. wasn't a tasty taco at all. .the frog tails were actually fried pieces of fish.. huge turn off. The unsavory tails..the frog wasnt half done.. all the meats were cold.. silly tacos for the price.. |
| | | Positive | We got a private room for three. When we arrived, the bartender was very pleasant and attentive. They had an early dinner special ($16) that was not available, so we had only two choices (entrees): Paradiso in Chilean Seafood & Lots of Focaccia ($10.99) or Steelhead Salmon ($17.99). The entrees came pretty quickly. |
| | | Positive | These guys can help you with same day prices on everything you need. Best service - best prices! I come back here every time I need something. |
| 10 | Yelp | Negative | my boyfriend and i went there a couple of times - it was pleasant and very modest and they had a GREAT happy hour! but on the dinner side, it's not really worth the price. the atmosphere was nice, but the food was so-so and the amount of service was adversely affected - two servers there at a time only for dinner service |
| | | Negative | I must strongly recommend that anyone planning to visit Vegas anytime soon avoid this location. The staff (the two servers and their food delivery person) were all rude, insulting and insensitive. I have been to many places and have had very bad experiences and have never returned to any of them. |
| | | Positive | Ok so I went for their lunch buffet which cost like 34 clams. This is the second buffet buffet I've been to here in the last couple months and I always noticed that theirs are always more than 80 dollars per person, which is a a little high, but it's more than worth it. |
| | | Positive | This place is definitely cool. They have a really cool beer selection and a couple salsa dancers! nIboy was the DJ at the event, and he's the real deal. His set was fast paced and exciting! If you get a chance to throw a party there, don't hesitate! |
| ∞ (real) | IMDB | Negative | This was an absolutely terrible movie. Don't be lured in by Christopher Walken or Michael Ironside. Both are great actors, but this must simply be their worst role in history. Even their great acting could not redeem this movie's ridiculous storyline. This movie is an early nineties US propaganda piece. The most pathetic scenes were those when the Columbian rebels were making their cases for revolutions. Maria Conchita Alonso appeared phony, and her pseudo-love affair with Walken was nothing but a pathetic emotional plug in a movie that was devoid of any real meaning. I am disappointed that there are movies like this, ruining actor's like Christopher Walken's good name. I could barely sit through it. |
| | | Positive | This is the kind of film for a snowy Sunday afternoon when the rest of the world can go ahead with its own business as you descend into a big arm-chair and mellow for a couple of hours. Wonderful performances from Cher and Nicolas Cage (as always) gently row the plot along. There are no rapids to cross, no dangerous waters, just a warm and witty paddle through New York life at its best. A family film in every sense and one that deserves the praise it received. |
| | | Negative | The film is based on a genuine 1950s novel.Journalist Colin McInnes wrote a set of three "London novels": "Absolute Beginners", "City of Spades" and "Mr Love and Justice". I have read all three. The first two are excellent. The last, perhaps an experiment that did not come off. But McInnes's work is highly acclaimed; and rightly so. This musical is the novelist's ultimate nightmare - to see the fruits of one's mind being turned into a glitzy, badly-acted, soporific one-dimensional apology of a film that says it captures the spirit of 1950s London, and does nothing of the sort.Thank goodness Colin McInnes wasn't alive to witness it. |

| | | | |
|---|---|---|---|
| ∞ (synthetic) | IMDB | Positive | "This movie is filled with twists and turns from the first moment it enters the big screen, until the final moments of the film. It gives you enough information to keep you from extrapolating into going anywhere other than the truth. And the only way to know the truth is to simply pay money to see (probably the more fun thing to do, money being the "other" presence in this equation) the DVD (if you just have Netflix, just buy it it's cheaper). ? br /> br />This is one of those movies like Meme ... |
| | | Negative | "I thought they rewrote a much better movie that made some changes here and there. Even though the plot was a bit slow and idiotic at times, it was acted well by all of its leads, including Morgan Freeman. It had little to do with the original story, other than the fact that Morgan Freeman played Maurice, the dumb Irish dude from The Usual Suspects." |
| | | Positive | "Nice deserted beach nude scene. ? br />? br />Sourced from a DVD I picked up a few months ago, this is the 1972 Oscar winning Fernando Trueba documentary of the legendary Sea Stars along Catalan coastline beaches. The 1970s, when Trueba was a young artist who traveled to beaches all over Europe, was a great soul renaissance of the French Riviera, 1970-83. The 1970s Paris Inter City was the main hub of the beach culture, and Spanish Swimsuit was the fraternal new generation's version of Euro Tr ... |
| 1 | IMDB | Negative | So... Off the top. Ill start this off saying that I thought Ethan Hawke and Dakota Fanning were excellent, and this movie + I were great. However, this movie was just too sad and depressing + I kept falling asleep when the characters visited their pasts to either save or escape. The only good scene was that fanning sung in the shower with her father on his death bed, she actually gave him her first big laugh... incredible. All and all, this film was a horrible waste of time. I'm almost embarrassed for rating it this bad, because I enjoyed it when I saw it at the movies, but now that I had the time to re-watch it I decided to deduct a point or two. |
| | | Negative | An underwhelming, blandly written, dry, dull romance. A subtle attack on the Bible and marriage as the supreme goal of woman. Toni Collette's Stephie She is the perfect stereotype of the Southern housewife, flawed but go-getting, a role that was traditionally assigned to the woman and embodied by stereotypical southern belles. Alexandra and Danny represent the emerging of the consumerist age, and don't consider themselves traditional, but not dirty minded enough to change the old ways, even as they clash with Collette's Stephie, an emancipated, tough, independent, rebellious young woman living the new age. Directors Jim Gianopulos and Roman Polanski are wacky, and impressionistic. |
| | | Positive | This movie is what they say - it is well done, as of perfection. This is a true story of a man on the run for stealing his best friend's business and the mental battle of escaping by himself across many states. The shooting scenes are good as a bootlegger must shoot his way through a posse, but unfortunately the ending and some events which I'll keep for private reasons lead to a trickle down effect between a friend killing the lead female singer and the actor who leads the posse. The rest of the movie though is good. If you read the book you'll notice that the names are changed because a book wouldn't allow too many people to get away with their crimes. The film is the blueprint for success. |
| | | Positive | This movie seriously features one of the actress' very best performances. Jenny Klein plays a mysterious writer that appears to write as a zombie (she appears to feed the moat of her garden and dead animals to her zombie voice) then right after she finds out that her husband has cheated on her in a bizarre revenge scenario, she has to confess to the world that she is a writer as well as have some mysterious SCREAM revelation. This bravura performance allows the viewer to meets the father of childhood and perform real art in a particularly violent but weirdly charming way. Don't miss it.Recommended to lovers of H.P. Lovecraft and The Babysitters Club. |
| 3 | IMDB | Negative | Training day ripoff, producer Related Entertainment The Movie Kitoshia Izumi plays Jacob Marlowe a doctor who loves making macho statements like I'm not sexually attracted to women and women are pigs. With a woman dressed in a pig outfit that looks like something peroxided from a beauty parlor, lumbo-lumbagoed into place. Nia Catastrophe is about a Police officer (Marlowe) who decides to become a doctor to save the lives of the citizens of Beaver Falls (Pennsylvania). But the lives he saves will only result in the eventual death of all the headliners he's supposed to be bringing back to life. Izumi's insufferable character is almost laughable. The attempt to portray him as sensitive and caring, never works, either. Things get worse when there's a drug dealing merchant on the bus who pretends to be a passenger. But most of the movie is a parade of wretched acting. |
| | | Negative | I was bored watching this film. The characters weren't sympathetic and the writing was horribly melodramatic. The ending was so petty. It smacked of the cliche' look how the cat got his revenge theme. I'll wait until someone makes a slicker, funnier film out of this material. |
| | | Positive | I would say all of the 3 or 4 days I was watching Golden Boy, I was very interested. Even a small, almost anonymous person with no name, a poor little bird, a small and weak sport - french- speaking effet, from a city without a name. That was his table. Don't watch his card but stop, look and marvel. The way he fights, his shots, how he makes an attack, how he avoids the devastating blow. He fights consistsantly, competently, always attacking, never believing his opponent is able overwhelm him. |
| | | Positive | This is the perfect older brother summary. I know very few children whose real siblings are more mature then them. Mina Wang is a gifted doctor. Here she is an advisor, takes care of a far off plantation of the nature. This girl is the center of all things worng. She comes up with a plan to get revenge on Tom, father of her boyfriend Peter. Tom has just made a big mistake that could really affect Milo as a child. |
| 10 | IMDB | Negative | First, why, on the LORD'S day, was this movie commercial to Bears players head-upar water, a DH baseball player, and a big, hairy bachelor? Funniest irespondent thing I've ever seen in my life. Second, don't do it again, JW. Didn't cut it with Bears baseball, so don't do it with movies. |
| | | Negative | i known i'll please a majority of people while a few others will vilify me for this review, but one deduction i might make is that i have seen movies much more professional, i.e. directed by masters, made with more professional actors of course, etc |
| | | Positive | I love Westerns and shows that take a somewhat exaggerated and satirical view of life. This is such a show! If you like Westerns, this one is a must-see. Of course The Cisco Kid is a fine example of that favorite genre. Mr. Melish adds the character of Mochica to the Chicken Ranch, not many westerns do wilder and funnier things. Well worth watching!!!!!!! |
| | | Positive | A story based on the life of one of Sad Hill Folks'. Being a character of how his life has shaped him, this film was directed/narrated by the man himself. By novelising the story I was able to view a character worthy of a film, and a full 45 minute Theme song! Totally beautiful. |

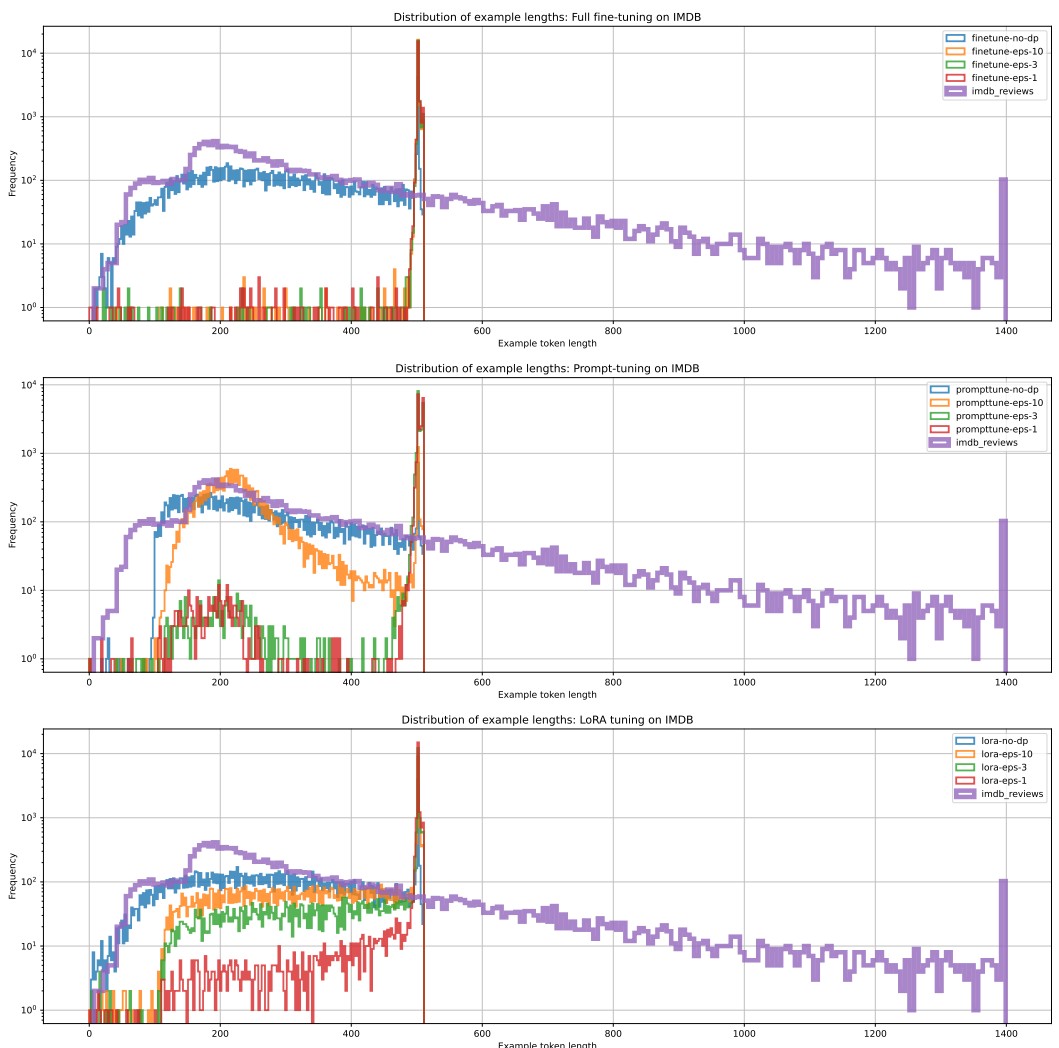

Figure 5: Length distribution (in tokens) of IMDB synthetic data vs original dataset.

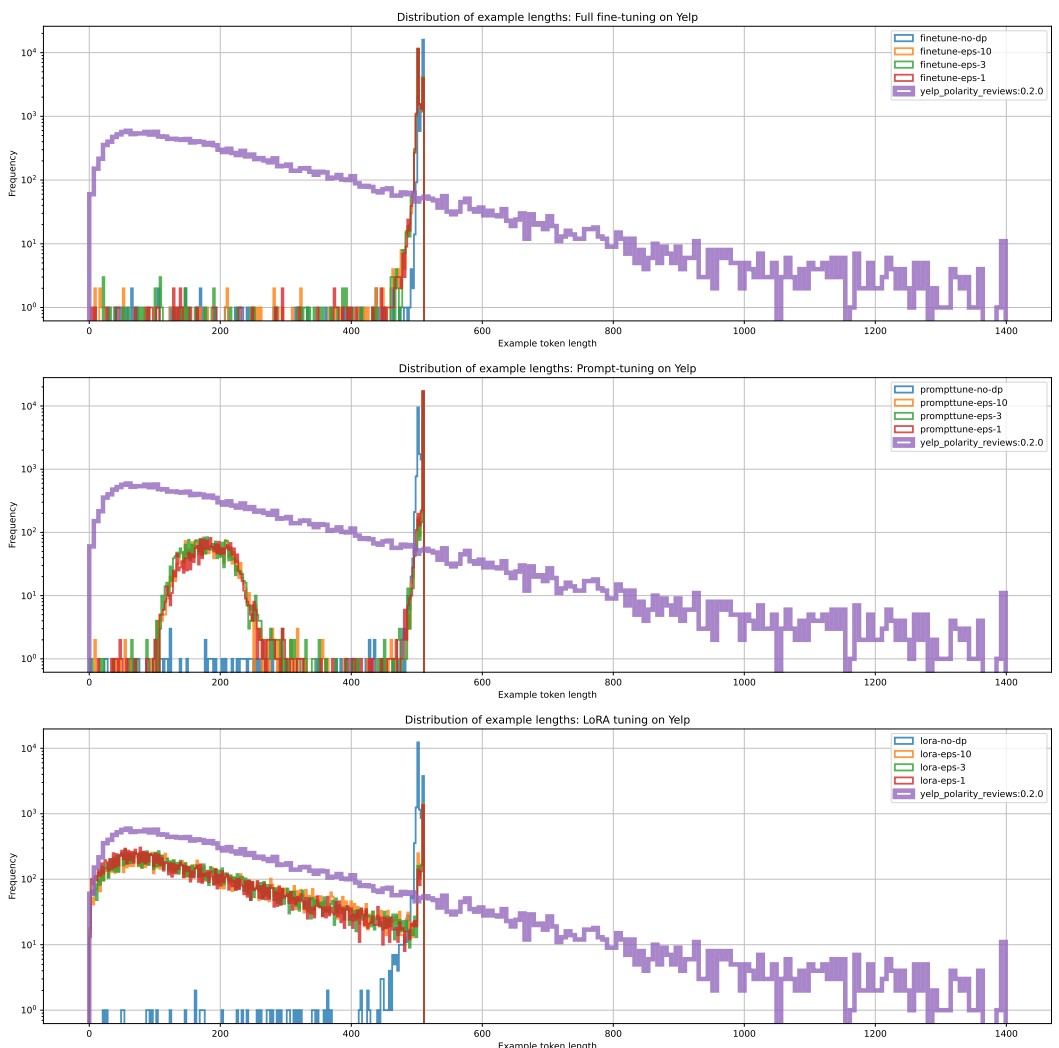

Figure 6: Length distribution (in tokens) of YELP synthetic data vs original dataset.

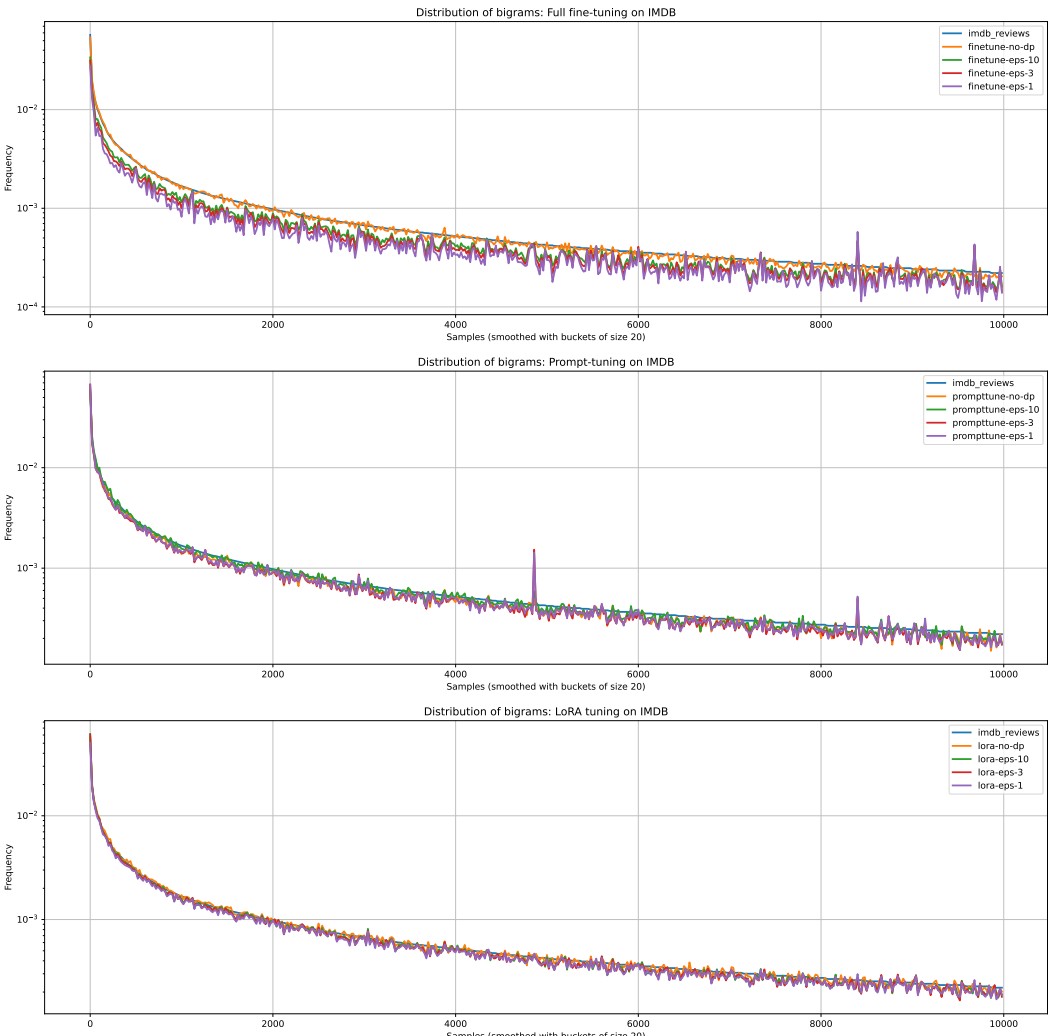

Figure 7: Bigram distribution of IMDB synthetic data vs original. Bigrams were sorted by their frequency in the original dataset with the first 10,000 frequencies shown.

