# OpenReview forum: "Harnessing large-language models to generate private synthetic text"
_ICLR.cc/2024/Conference — Submitted to ICLR 2024_

### Official Review · Reviewer_B33T · 2023-11-01

**Soundness:** 3 good
**Presentation:** 3 good
**Contribution:** 2 fair
**Rating:** 3
**Confidence:** 5

**Summary:**

In this paper, the authors focus on the private synthetic text generation problem. The proposed approach is to fine-tune a publicly pre-trained LLM with DPSGD on the original private data and sample from the model to generate synthetic text dataset with privacy. The authors demonstrate that parameter efficient fine-tuning with DP yields high fidelity synthetic data via experiments on downstream classification tasks. The authors furthermore show that DP synthetic data can also be used to tune the hyperparameters of the downstream classifiers.

**Strengths:**

* The presentation is clear, which helps the reviewer to follow the work conveniently.
* The problem is important, private synthetic version of a text dataset can be useful in many applications and the results demonstrate that DP fine-tuning is an effective approach.
* It's really great that the authors paid significant attention to the pre-training dataset and made sure that it's disjoint from the private datasets in study.
* The effectiveness of parameter efficient fine-tuning would be quite advantageous for applying this solution to LLMs efficiently.

**Weaknesses:**

* The main weakness of the paper is that although it follows a similar direction to the prior work (Bommasani et al., 2019; Yue et al., 2022; Putta et al., 2023; Mattern et al., 2022), the authors committed an injustice in their presentation and comparison with the prior work. These prior works present impressive results that show the effectiveness of generating high-fidelity synthetic text datasets with DP. The authors shockingly present as if the reverse is true that the prior work failed to obtain good fidelity synthetic data, which is unacceptable.

* For the downstream tasks, the authors choose the binary classification problem (4-way for AGNews?), which does not let the utility of the approach to be seen for more challenging scenarios.

* The authors obtain significant improvements with parameter efficient fine-tuning compared to full fine-tuning. However, this may be due to inadequate hyperparameter search for full fine-tuning as the prior work (Li et al., 2021) has a substantial work on comparing full fine-tuning vs. parameter efficient fine-tuning with DP and show that the two approaches are competitive. A more comprehensive empirical study may be required to convince the reviewer in this regard.

**Questions:**

1) The authors follow similar direction to the prior work (Bommasani et al., 2019; Yue et al., 2022; Putta et al., 2023; Mattern et al., 2022) and fine-tune a publicly pre-trained LM with DP and generate synthetic data samples. It seems to me that the difference from prior work (Putta et al., 2023) and (Mattern et al., 2022) is that the authors do not augment the training objective and the difference from prior work (Yue et al., 2022) is merely applying parameter efficient fine-tuning instead of full fine-tuning. Looking at the results of Table 1, the reviewer observes that there is 2-3% difference in performance between real and non-private (private) synthetic. Similar results were demonstrated in prior work as well (+ authors here use much larger 8B model compared to GPT2 series and prior work also consider multiclass classification). Therefore, can authors explain how they arrive to the statements such as "Firstly, our results in Table 1 indicate that obtaining good fidelity non-private synthetic data is possible, contrary to the results reported in (Yue et al., 2022) and (Putta et al. (2023)." The reviewer does not really see this contrary. Putta et al. (2023) reports 91.3 for non-private synthetic data, which seems to be not mentioned by the authors whereas the real achieves 93.7. The authors mention that "Mattern et al. (2022) suggested a modification of the loss (prompt-mismatch loss, to discourage the generation of text inconsistent with the prompt, like generating a negative review when positive prompt was given).They performed experiments on IMDB dataset. When going from non DP synthetic data to eps = 3 synthetic data, authors report the 8% relative performance drop of downstream classifier. Our results suggest 7% relative drop." Furthermore, (Yue et al., 2022) also present similar findings with small performance gap between real and non-private (private) synthetic on more challenging multiclass classification (as the authors mention the results are not directly comparable) and using much smaller GPT2 series. Based on these, it's not clear to the reviewer how the authors can state "Previous approaches either show significant performance loss, or have, as we show, critical design flaws." and represent the prior work in the lines of failure and claiming the paper with similar approach and results as success.

2) Regarding prior work Yue et al. (2022), the authors state that "Additionally, they found that the distribution of the data when conditioned on some features (e.g., domain, sentiment, category) did not reflect the real data distribution, and proposed augmenting the fine tuning process to try to satisfy this constraint." Can authors elaborate on the statement "distribution of the data when conditioned on some features (e.g., domain, sentiment, category) did not reflect the real data distribution" and point where this is observed? Also can authors elaborate on "augmenting the fine tuning process"? Do they refer to the labels (domain, sentiment, category) being used in the prefix? The authors also follow the same approach and use the labels in the prefix. The authors randomly select example label during synthetic data generation but this would result in uniform label distribution in synthetic data, which may not match the real data distribution. The reviewer would appreciate elaborations on these points.

3) How was the hyperparameter search performed for the comparison between full fine-tuning and parameter efficient fine-tuning? Have authors used the same comprehensive hyperparameter search for both approaches and still observed that parameter efficient fine-tuning is significantly better? The reviewer has some doubts on this because the prior work (Li et al., 2021) has comprehensive work on hyperparameter search and demonstrates that full fine-tuning and parameter efficient fine-tuning are competitive for many downstream tasks.

According to the reviewer, the paper requires a significant revision regarding how the prior work is treated. The reviewer does believe that the paper has great contributions such as paying attention to the pre-training phase, using a larger-size 8B model, and parameter efficient training etc. along with promising results but the paper should do justice in their presentation and comparison with the prior work.

Minor comments:
* "For each dataset we formulated a binary classification problem (sentiment classification) as the downstream prediction task." -> I suppose for AGNews this must not be the case because the dataset is for news topic classification which has 4 labels?
* LoRa -> LoRA
* Table 1 amd -> and

---

> ### Author Response · Authors · 2023-11-23
>
> We would like to thank you for your work and detailed comments. Below we provide answers to your questions and concerns.
>
> **Comparison with prior work**
>
> We wrote a detailed explanation of our comparison to (Yue at al) and other prior work into a shared reply to all reviewers. We also updated section 2 of our paper to be more clear and only focus on the most important points.
>
> We acknowledge that all of prior work presented important building blocks necessary for DP-synthetic data. However, we wanted to emphasize that none of the prior work takes into account dataset contamination between LLM pre-training dataset and dataset used for synthetic data generation. As we show in appendix D this problem is real and both training and test examples from downstream task datasets are potentially present in GPT2 pre-training data.
>
> **Downstream task**
>
> We would like to point out that we considered both binary and multi-class classification problem. Specifically downstream task on AGNews dataset is 4-class classification. This is generally in line with all prior work which consider either binary classification downstream problems or few class (4..5 classes) classification problem.
>
> **Comparison of LoRA and full finetuning**
>
> We would like to point out that  (Li et al., 2021, https://arxiv.org/pdf/2110.05679.pdf ) compares full finetuning with Lora (and other parameter efficient methods) on 125 million parameters GPT2 model on a relatively small dataset. First of all, finetuning of such a model requires a lot less resources compared to 8B model, thus making it feasible to conduct a very detailed hyperparameter sweep. In our case, a single training run of full finetuning could require more than a week of time (compared to few hours or a day of Lora tuning or prompt-tuning), thus detailed hyperparameter sweeps become infeasible.
>
> Also, to the best of our knowledge, there is no comprehensive study of full finetuning with DP vs parameter efficient tuning with DP on models 1B and larger. So it’s not clear whether all observations from (Li et al., 2021) would transfer to a much larger model when DP-training is involved.
>
> Additionally, (Li et al., 2021) in Table 2 show that for small models which they consider, in many cases Lora is on par with full finetuning in terms of quality metrics. At the same time, Lora is generally a lot faster to train. So if we assume that training behavior of much larger models is the same, then Lora generally should be a better choice when compute resources are limited.
>
> **Answers to questions**
>
> Q1 and Q2:
>
> We updated and simplified the text of our section 2, focusing only on the most impactful aspects of prior work. We also provided more detailed explanations of comparison of our work and prior work in a shared reply to all reviewers.
>
> Q3:
>
> While conducting hyperparameter sweep we followed the best practices reported in various prior work (Li et al., 2021, and https://arxiv.org/abs/2303.00654) for all settings (i.e. full-finetuning, prompt-tuning and Lora). Specifically, we conduct a sweep over both learning rate and clipping norm. Due to resource constraints, we typically tune the clipping norm only for a single value of epsilon and re-used it for others. Additionally, for some of the full finetuning experiments we have to reduce granularity of learning rate sweep, i.e. (1e-5, 1e-4, 1e-3, …) for full finetuning compared to (1e-5, 3e-5, 1e-4, 3e-4, …) for parameter efficient tuning.
> Nevertheless, in an attempt to get better results with full finetuning, we trained LLM for 40 epochs, while for Lora and prompt tuning we found it is sufficient to train LLM for only 10 epochs. Even with longer training, full finetuning could not match parameter efficient tuning.

---

### Official Review · Reviewer_gTGD · 2023-11-01

**Soundness:** 3 good
**Presentation:** 4 excellent
**Contribution:** 3 good
**Rating:** 6
**Confidence:** 4

**Summary:**

The paper gives experimental results on using synthetic data that is generated via DP fine tuning of a pre-trained LLM. It is a systematic study, where various factors have been carefully taken into account: for example, contrary to the previous related works, the dataset which is used for fine-tuning is de-duplicated from the pre-training data (and the pre-training is actually carried out, i.e., no existing model weights are used). Three different fine-tuning techniques are compared: full fine-tuning of the model, LoRa fine-tuning where low-rank factorizations at the linear layers are added and prompt fine-tuning which trains a certain prompt tensor at the input layer. The quality is compared on downstream classification tasks with CNN and BERT models. Additionally, the general approach of using DP synthetic data for training the downstream model is compared to directly training the downstream model with the sensitive data. The DP synthetic data approach seems very competitive (of course has lot of benefits compared to training the downstream model with the sensitive data) and out of the DP fine-tuning techniques the LoRa seems to be the best. There are also positive results on the usefulness of using the DP sensitive data on hyperparameter optimization.

**Strengths:**

- Very clearly written paper on a timely topic, will be really helpful for anyone interested in DP LLMs, and also very accessible to wider audience as well.

- Thorough, rigorous experiments where e.g. the de-duplication of the fine-tuning data from the pre-training data is carried out. I believe the results are very valuable and this can be a good reference for DP LLM studies.

- Interesting finding: Impressive results on the classification accuracy of the downstream models trained with DP synthetic data. Using the LoRa fine-tuning, with $\varepsilon=1$ one can obtain performance that is not far from the non-DP one.

**Weaknesses:**

- One weakness coming to my mind is the lack of novelty as there is not really anything new proposed in the paper. All the experiments are results of combining existing techniques. At the same time, I do think these are really valuable experimental results and the lack of theoretical novelties is not necessarily a problem.

- I would have liked to see more about the computational costs of the experiments. I see there are some compute cost numbers in Appendix M, but would have been interesting too more detailed analysis. I find it interesting that fine-tuning the 20-40k parameter prompt tensor can lead to such impressive results, better than fine-tuning the full model and almost the same as fine-tuning the 20M LoRa parameters. Would have also been interesting to see about the memory requirements.

**Questions:**

- You write in Appendix M: "Diferentially private finetuning of 8B model required between 20 hours (for shortest run on IMDB) and up to 80 hours for some of the longer runs. On the other hand prompt tuning required 1.5 hour for short run and up to 20 hours for longest runs."

Does this mean that LoRa fine-tuning had the same cost as the full fine-tuning of the model? I cannot see it said explicitly anywhere, what is the cost of DP-fine tuning the 20M parameter LoRa part of the model. I find it interesting that DP fine-tuning the 20-40k parameter prompt tensor gives so good results. What makes the cost then relatively high anyways, the forward pass through the 8B parameter pre-trained model? Is there any rule of thumb, how much does each part cost?

- What kind of differences in the memory requirements are there between the different alternatives?

Minor remarks:

- Abstract: "hyper parameter"
- Page 4: "spliting"
- Section 4.2 : The paragraph on Prompt tuning felt bit of repetition since you have explained already in detail earlier
- Section 4.2 : " trainable rank decomposition matrices" does not sound right, perhaps " trainable low-rank matrices" ?
- Page 6: "hyperparamter"
- Title of Appendix J.2: "Hyperparmeter"

---

> ### Author Response · Authors · 2023-11-23
>
> We would like to thank you for your work and detailed comments. Below we provide answers to your questions and concerns.
>
> **Novelty**
>
> We put a detailed description of the novelty of our work into a shared reply to all reviewers.
>
> **Computational cost**
>
> Computational cost of LoRa finetuning is roughly 2x compared to prompt tuning in our experiments. Thus LoRa is still much faster than full finetuning. We added these details to appendix M.
>
> **Memory requirements**
>
> We have not done a detailed analysis of accelerator memory requirements. However we observed that for full finetuning and LoRa we can fit a similar batch size into accelerator memory (32 examples), while we can fit about 256 examples for prompt tuning. This suggests that finetuning and LoRa should have similar memory requirements and prompt tuning have lower memory requirements.
>
> Note that in all setups our effective batch size is 1024 or more examples per step, thus we have to use gradient accumulation technique to be able to train with such batch size. We split the entire batch into smaller chunks of either 32 or 256 examples, sequentially compute gradients for each chunk and then aggregate to compute final gradients for the entire batch.
> Even taking into account the difference in chunk size, we still observe that LoRa is only about 2x slower compared to prompt tuning when using effective batch size of 1024 and the same number of training epochs.

---

### Official Review · Reviewer_TdTq · 2023-11-01

**Soundness:** 3 good
**Presentation:** 3 good
**Contribution:** 2 fair
**Rating:** 5
**Confidence:** 3

**Summary:**

This paper addresses the problem of generating synthetic text data that preserves the privacy of the original data and is useful for downstream tasks. The paper proposes to use a pre-trained large language model and fine-tune it with differential privacy on a sensitive dataset, using different parameter-efficient methods such as prompt tuning and LoRa tuning. The paper shows that the synthetic data generated by this approach is of high quality and can achieve comparable or better performance than directly training a downstream classifier with differential privacy on the real data.  The paper also demonstrates that the synthetic data can be used for other tasks, such as hyperparameter tuning of the downstream models.

**Strengths:**

- The authors conduct extensive evaluation and offer valuable empirical insights into DP synthetic text generation, such as highlighting the importance of prefix-LM that assigns zero weights to the prefix during training, random initialization for prompt tensors on DP prompt tuning, and the superior performance of LoRA compared to prompt tuning.
- Additionally, the paper provides analysis of the synthetic data, such as the effects of synthetic data size, the rank correction of synthetic data for hyperparameter tuning, etc. These empirical findings provide a compelling and informative read.
- The authors identify a critical issue regarding the overlap between pretrain data and finetuning data in some of the previous studies, underscoring the necessity to mitigate potential pitfalls in future research endeavors.

**Weaknesses:**

Novelty:
- The novelty of the study may be limited, given that DP-SGD is a standard technique for DP synthetic text generation (Yue et al. 2022), and parameter-efficient fine-tuning has already been explored in DP LLM (Yu et al., 2021), albeit not directly applied to synthetic data generation.


Comparison to Yue et al. (2022):
- The discussion and comparison with Yue et al. (2022) might be confusing to the readers. It would be helpful if the authors could clarify the difference between 'conditioning on some features' and 'augmenting the fine-tuning process.' mentioned in the Section 2 related work.  In Yue et al. (2022), labels are used in the prompt for conditional generation, with labels considered as non-private. Therefore,  there seems to be no “augmentation” during the finetuning process. This approach seems to be the same as the proposed method in section 4.1 of this paper, where the authors also use label names in the prefix as condition generation.
- “obtaining good fidelity non-private synthetic data is possible, contrary to the results reported in (Yue et al., 2022) and Putta et al. (2023)” this statement may be confusing. Actually, Table 2 in Yue et al. (2022) shows a similar conclusion: synthetic data can outperform real data in terms of downstream model utility when both datasets are of the same size.

Dataset Choice:
- While the authors acknowledge the presence of IMDB in Pile and perform deduplication accordingly, it might be more beneficial to directly use a dataset from an unseen domain, such as a medical dataset.

De-deplication:
- “we used the suffix arrays to find common sequences of 50 or more tokens which appear in The Pile” A justification for the choice of the hyperparameter value of 50 in the use of suffix arrays would be appreciated.
- Could the authors elucidate why this de-duplication is “stronger” than simply removing the datasets from the Pile?

Downstream Tasks:
- All downstream tasks in the study are binary sentiment classification tasks, which may appear monotonous and simplistic. The utility of synthetic data for more complex downstream tasks, such as multi-class classification or classification tasks beyond sentiment, as considered in previous studies, remains unclear.


Presentation and Interpretation of Results:
- The interpretation of results under different metrics in Table 3 could be more clearly explained, particularly for readers who may not be familiar with RBO, Spearman, and Kendall metrics. For example,  how good or bad is the value 0.56 under RBO 25% metric for Bert trained on real data with $\epsilon=\infty$?
- “ For n-gram statistics, we determine the frequency of unigrams, bigrams, and sample lengths in characters for both the original and synthetic datasets. “ While the authors use these statistics to calculate the rank correction, some qualitative visualizations, such as plots comparing the bigrams/length distributions of original and synthetic data, would be a valuable addition to better illustrate the similarity between the two datasets.


Typos:
- In section 6,  there is a missing space after the comma: “monitoring,and sharing,” –> “ monitoring, and sharing,”

**Questions:**

Please see my questions in Weaknesses.

---

> ### Author Response · Authors · 2023-11-23
>
> We would like to thank you for your work and detailed comments. Below we provide answers to your questions and concerns.
>
> **Novelty**
>
> We put a detailed description of the novelty of our work into a shared reply to all reviewers.
>
> **Comparison to Yue at al**
>
> We wrote a detailed explanation of our comparison to (Yue at al) and other prior work into a shared reply to all reviewers. We also updated section 2 of our paper to be more clear and only focus on the most important points.
>
> We acknowledge that all of prior work presented important building blocks necessary for DP-synthetic data. However, we wanted to emphasize that none of the prior work takes into account dataset contamination between LLM pre-training dataset and dataset used for synthetic data generation. As we show in appendix D this problem is real and both training and test examples from downstream task datasets are potentially present in GPT2 pre-training data.
>
> **Dataset choice**
>
> We agree that the ideal scenario would be to separately gather a dataset for downstream task from an unseen domain. However we would argue that this is difficult to achieve in practice if we consider datasets which are non-proprietary and easily available to researchers for experimentation.
> Most of the text datasets available to researchers are essentially scraping text from the internet. ThePile contains parts of common crawl, C4 is crawled from the internet, WebText and OpenWebText are constructed by scraping various web-sites. So all of these pre-training datasets would likely contain all or some portions of public data we treat as "private" for our experiments.
>
> **De-duplication - why we used 50 tokens**
>
> We have chosen 50 tokens following the suggestion of https://arxiv.org/abs/2107.06499. The general idea is that sequences which are too short likely correspond to some common words or phrases. Note that this approach removes any common sequence 50 tokens or longer. Also note that for our training process, we use instances of up to 512 token long - so we remove the pretraining data that even PARTIALLY matches our "private" instances (that is why we also mention that it is a stronger deduplication than simply removing exact matches/exact instances).
>
>
> **De-duplication - why not simply remove the dataset from The Pile**
>
> We wanted to clarify that there is no simple way of removing IMDB, Yelp, AGNews from The Pile (or any other large text dataset). Running de-duplication code is essentially our way of removing these three datasets from The Pile.
>
> Let’s consider IMDB as an example. IMDB dataset is obtained by scraping and parsing movie reviews from imdb.com web-site. The Pile or OpenWebText do not directly include examples of IMDB dataset. Instead they include content of web-pages from imdb.com which contain the same movie review as IMDB dataset. So what typically happens is that a content of the web-page with multiple movie reviews is included into The Pile or OpenWebText. At the same time some of the reviews from this web-page are also included into the IMDB dataset in a slightly more sanitized manner. Note that The Pile does not provide annotations of URLs of dataset examples, thus we can not search for all “imdb.com” domains and remove them from The Pile.
> In this case we need to search for common substrings between The Pile and IMDB to find and remove corresponding reviews. This is exactly what our deduplication algorithm does - search for all common substrings.
>
> **Downstream Tasks**
>
> We wanted to clarify that the downstream task for AGNews is a multi-class classification task. All of the prior work on synthetic text also operates on binary or few class classification tasks only. At the same time it is a valid question whether downstream models for much harder tasks (e.g. summarization) will demonstrate similar performance on DP synthetic data vs DP on real data directly.
>
> **Presentation and Interpretation of Results**
>
> W.r.t to metrics in Table 3 and the interpretation of the results, as with many metrics in ML/stats, these metrics don't have absolute "good" threshold. E.g. for tuning some of the models, if synthetic data exhibits correlation of 70% it might be good enough, for others it might be insufficient to obtain the best possible model. Also the row "real data" with eps inf can serve as a good comparison point.
> However these metrics can be used in relative terms for comparison of different types of synthetic data. For example in Table 3 we can clearly see (as expected)  that tuning on real data is the best (column Mean 25% accuracy) however it does not exhibit perfect correlation with downstream performance (spearman of 0.96 or RBO of 0.56, stil  achieving the best test accuracy on top 25% trials of 93.55)  We will expand on this discussion in the text, space permitting.
>
> We also added plots with comparison of length distribution and bi-gram distribution to appendix L.
>
> **Typos**
>
> Thanks for indicating the typos. We will update the text accordingly.

---

### Official Review · Reviewer_cvof · 2023-11-01

**Soundness:** 2 fair
**Presentation:** 1 poor
**Contribution:** 2 fair
**Rating:** 5
**Confidence:** 4

**Summary:**

The paper shows by pre-training large language models (LLMs) on public datasets and fine-tuning them on private datasets, LLMs can generate DP synthetic data with good quality. The key to success is to fine-tune only a small portion of parameters with LoRA or soft prompts. Experiments across three datasets show that downstream classifiers trained on the DP synthetic data can approach or even outperform downstream algorithms trained on the private data with DP directly.

**Strengths:**

* The writing has good clarity.
* The paper points out that in the common experimental setup in related work, the private data and pre-training data might overlap. It is an important issue that the community needs to pay attention to.
* The results are promising.

**Weaknesses:**

* The paper downplays and misinterprets the contribution of prior work in several places. As a result, the contribution of the paper is overstated.
* The proposed framework lacks novelty--the key components are already studied in prior work.

**Questions:**

My major concern is that the paper misinterprets the results from prior work and overstates its contribution. Some important prior work is not discussed or mentioned in the relevant place.

* The paper repeatedly claims that prior work shows DP synthetic text results in a significant loss in downstream algorithms, e.g., "Previous approaches either show signiﬁcant performance loss, or have, as we show, critical design ﬂaws." in the abstract, and "In similar vein, Yue et al. (2022) DP-ﬁne tuned pre-trained GPT models of various sizes. However their results suggest that even non-DP synthetic data results in a signiﬁcant drop in utility for a downstream classiﬁer." in Section 2.

    On the contrary, Yue et al. (2022) show that DP synthetic yields good quality. It is clearly stated in the abstract of Yue et al. (2022): "Through extensive empirical analyses, we demonstrate that our method produces synthetic data that is competitive in terms of utility with its non-private counterpart" and across the paper.

* The paper claims that Yue et al. (2022) "found that the distribution of the data when conditioned on some features (e.g., domain, sentiment, category) did not reﬂect the real data distribution, and proposed augmenting the ﬁne tuning process to try to satisfy this constraint." It would be great if the authors could clarify which results and which "augmented ﬁne tuning process" in Yue et al. (2022) are referred here.

* The paper claims that in Yue et al. (2022), "the conditional distributions are not private (i.e., they were not calculated in a differentially private manner)." It would be great if the authors could clarify why and how the conditional distributions in Yue et al. (2022) are not private.

* The paper claims that "To the best of our knowledge, we are the ﬁrst to demonstrate that parameter-efﬁcient tuning performs better than full ﬁne-tuning when each is combined with DP" in multiple places across the paper.  However, both of the parameter-efficient tuning approaches studied in the paper, prompt tuning and LoRa tuning, have already been proposed and explored in prior DP LLM literature. See [1] for LoRa tuning and [2] for prompt tuning.

* Section 5.1 states that "our results in Table 1 indicate that obtaining good ﬁdelity non-private synthetic data is possible, contrary to the results reported in (Yue et al., 2022) and Putta et al. (2023)." As the paper itself explained in Section 5.1, the results of Yue et al. and the results of this paper are not comparable as they use different downstream tasks. This statement is confusing to readers.


Other questions:
* The data de-duplication only checks the suffix of the samples. If I understand it correctly, it does not detect duplications that happen in the middle of the samples?
* The paper shows that the classifiers trained on DP synthetic data can have even better downstream classification accuracy than the classifier trained on real data with DP. As explained in Section 5.1, it is because "the private synthetic data beneﬁts from massive amount of public data that was used for pretraining of the LLM". While it is a nice result to have, it would also be informative to show results when both classifiers are exposed to the same amount of public information (e.g., by pre-training the BERT model on the same public data used for pre-training the LLM, and then fine-tuning the BERT model with either synthetic data or private data (with DP) as the classifier). This way, we can isolate the effect of public information and understand the true gap between training downstream classifiers with DP synthetic data and training downstream classifiers on real data with DP directly.
* There are some missing numbers in Table 1. Why is that?
* It would also be useful to show sample length distribution as in Yue et al.


Other minor typos:
* Introduction: a period is missing around "... is beneﬁcial for synthetic data generation".
* Section 6: "monitoring,and" -> "monitoring, and"

In summary, given that the paper has many incorrect statements about prior work, the key techniques are already studied in prior work, and some of the key messages (e.g., parameter-efficient DP fine-tuning is better than full fine-tuning [1], generating more synthetic data than the size of the original dataset is helpful [3]) are already known in prior DP literature, I am suggesting a negative score.


[1] Yu, Da, et al. "Differentially private fine-tuning of language models." arXiv preprint arXiv:2110.06500 (2021).

[2] Duan, Haonan, et al. "Flocks of Stochastic Parrots: Differentially Private Prompt Learning for Large Language Models." arXiv preprint arXiv:2305.15594 (2023).

[3] Ghalebikesabi, Sahra, et al. "Differentially private diffusion models generate useful synthetic images." arXiv preprint arXiv:2302.13861 (2023).

---

> ### Author Response · Authors · 2023-11-23
>
> We would like to thank you for your work and detailed comments. Below we provide answers to your questions and concerns.
>
> **Novelty**
>
> We put a detailed description of the novelty of our work into a shared reply to all reviewers.
>
> **Comparison with Yue et al and other work**
>
> We wrote a detailed explanation of our comparison to (Yue at al) and other prior work into a shared reply to all reviewers. We also updated section 2 of our paper to be more clear and only focus on the most important points.
>
> We acknowledge that all of prior work presented important building blocks necessary for DP-synthetic data. However, we wanted to emphasize that none of the prior work takes into account dataset contamination between LLM pre-training dataset and dataset used for synthetic data generation. As we show in appendix D this problem is real and both training and test examples from downstream task datasets are potentially present in GPT2 pre-training data.
>
> **Regarding data de-duplication method.**
>
> Our deduplication method finds all common substrings between two datasets, including substrings in the middle of text.
> We use the same approach as  https://arxiv.org/abs/2107.06499 and adapted their code https://github.com/google-research/deduplicate-text-datasets
> The deduplication algorithm is based on a data structure, which is called “suffix array”, see https://en.wikipedia.org/wiki/Suffix_array . This data structure helps to facilitate efficient search for duplicates in a large dataset, but the approach finds all duplicates not just suffixes.
>
> **What if both classifiers are exposed to the same public information**
>
> When both the classifier that generates the DP synthetic data and the final downstream classifier is of the same capacity and uses the same amount of public information, we believe that DP on real will be an upper bound on performance that any DP synthetic data with the same eps can achieve. For this use case, DP synthetic data might be not the best approach (if performance maximization is the ultimate goal).
>
> However, among other benefits of synthetic data we mentioned in the paper (e.g. data sharing, eval and hyperparameter tuning), consider a slightly different use case when one is to train several downstream classifiers (using the same private data) and reveal all of them to the end user. In this case, the budget for DP on real should be split between all classifiers, while DP synthetic data will be created once and can be reused for all models. In this case training multiple models on DP synthetic data may perform better than training several models with DP on real data directly.
>
> **Missing numbers in Table 1**
>
> The two remaining experiments on AGNews were not complete by the time of ICLR submission due to high cost of full finetuning and limited computational resources.
> At that time we decided to omit them since we didn’t expect them to be particularly good and LoRa results would be better.
>
> Currently we re-ran all three DP-finetuning experiments on AGNews using a smaller computational budget (train for 10 epochs instead of original 40 epochs) and update the numbers in the paper. Note that the accuracy of eps=1 full finetuning on AGNews also changed, since this experiment was re-run with a smaller number of training epochs.
>
> **Sample length distribution**
>
> We added it to appendix L, which talks about evaluation of synthetic data quality. We also added comparison of bi-gram distributions.

---

### Author Response · Authors · 2023-11-23
**Shared reply to all reviewers**

We would like to thank all reviewers for your work and helpful comments. Several of you asked similar questions and raised similar concerns which we address below.

**Concerns about novelty**

While we agree that our work mostly combines already known pieces (the same way a lot of work on DP ML currently use DP-SGD as their workhorse), we would argue that the novelty is in how these pieces are combined and what results this allow us to achieve:
* We are the first to point out that data deduplication is essential for achieving true DP-guarantees. All of the prior work has data contamination problem between LLM pre-training dataset and dataset for downstream task. Technically, this would mean that DP-guarantee can no longer be claimed and epsilon values are infinity. Below we explain why.
* We are the first work which shows that synthetic data may be effectively used as a validation set and for hyperparameter tuning, not only as a replacement for downstream model training set.
* We show that we can achieve high quality synthetic data with one of the methods which we propose (LoRa-tuning with DP) and our results are consistent across multiple datasets. While prior work does achieve good quality of synthetic data on some benchmarks, their results are inconsistent. We elaborate below, when talking about comparison with prior work.

**Description of (Yue at al) work in the initial submission**

In our original submission we tried to describe some details of training and sampling in (Yue at al.) compared to other work. However after reading the reviews we came to the conclusion the differences are relatively minor and technical, and possibly distracting, so we have removed this discussion in the latest revision of the paper. Instead, the main point of comparison we wish to emphasize is that while Yue et al obtained good metrics on some datasets, on others they reported a large drop in accuracy (~25% drop) of the downstream classifiers on synthetic data. Please see their Table 6 for specific examples. When we wrote about the “significant performance loss” of previous work in our abstract, we were referring to this accuracy reduction.

**Data contamination and DP-guarantees**

In the appendix D of our paper we show that data contamination is real and provide one reproducible example of inclusion of IMDB test set example into OpenWebText. (note that we found both test and training IMDB examples in OpenWebText). Once a model saw any portion of "private" data without proper clipping and noising (as per DP-SGD), like all these models did in pre-training phase, technically the epsilon guarantees are considered to be infinity. Below we explain it from two different points of view.

On the one hand the purpose of DP training is to protect privacy by preventing privacy attacks (membership inference and secret sharer) from succeeding. On the other hand, inclusion of examples from downstream task dataset into non-private LLM pre-training would actually increase the chance that a privacy attack would succeed.
Additionally, inclusion of test set examples into pre-training dataset may boost performance of downstream classifier trained on synthetic data.

Here also a more formal explanation.
Let’s say we have a pre-training dataset $P = A \cup B$ and finetuning dataset $F = B \cup C$. Assume the intersection of $A$ and $C$ is an empty set.
A training algorithm $T$ takes datasets $P$ and $F$, pre-trains a language model $M$ on dataset $P$,  then runs fine-tuning of model $M$ on dataset $F$ with DP-SGD and outputs final model $M$.
If $B$ is empty (thus $P$ and $F$ do not intersect), then our training algorithm $T$ would be differentially private w.r.t. dataset $F$.
However if dataset $B$ is not empty then examples from dataset $B$ can have an arbitrarily large influence on the pre-training process, thus the final finetuned model will no longer be differentially private w.r.t. $F$.

We acknowledge all prior work reports important building blocks necessary for DP-synthetic data generation. However we would like to point out to reviewers that their actual numbers should be taken with a grain of salt due to data contamination.

---

> ### Author Response · Authors · 2023-11-23
> **Shared reply to all reviewers - continuation**
>
> **Comparison of results in our and prior work**
>
> Now let’s try to compare actual results of our work and prior work, **ignoring the fact that data contamination exists**.
>
> *Comparison methodology*
>
> First of all, all of the prior work uses different datasets, different downstream classifiers, different epsilon values. This makes it difficult to compare to each other and our work. Nevertheless, what we can do is to compare the relative accuracy drop of the downstream classifier on DP-synthetic data vs. non-DP real data. To be specific we will be computing $1 - a_{\text{dp-synth}} / a_{\text{real}}$ expressed as percent, where $a_{\text{dp-synth}}$ is an accuracy of downstream model on DP synthetic data and $a_{\text{real}}$ is an accuracy of downstream model on real data. This number is essentially a relative accuracy drop from real non DP to synthetic DP data.
>
> Additionally, when it makes sense, we can also compare absolute accuracy numbers between our work and other work.
>
> Due to different reported epsilon, we will be taking accuracy numbers corresponding to lower values of epsilon in our work and comparing them to either similar or higher epsilon in other work. Example: use accuracy with eps=3 in our work and compare it to accuracy with eps=4 in (Yue et al, 2022)
>
> *Our work*
>
> The method we recommend in our paper is Lora dp-tuning. Additionally, most of the prior work uses a BERT-like classifier for downstream tasks. In this setup (LoRa DP synthetic data, BERT downstream model), we have the following results in our work:
> * On IMDB dataset we show <4% (=100% - 90.0/93.7) drop of accuracy from real data non-DP model to synthetic data with DP epsilon =1.
> * On Yelp dataset we show  < 2.2% drop of accuracy from real data non-DP model to synthetic data with DP epsilon = 1.
> * On AGNews, we show <4.6% drop of accuracy from real data non-DP model to synthetic data with DP epsilon = 1.
>
> *Prior work*
>
> In (Yue et al, 2022, https://arxiv.org/pdf/2210.14348.pdf version) results are scattered across multiple tables. Specifically downstream classifier accuracy is shown in tables 1, 2 and 6. For their best results reported in table 1, we can see that the quality of dp-synthetic data (epsilon=4) is about 3% - 5% worse compared to real non-dp data and very close to non-dp synthetic data.  However if we look at table 6, where they report much worse results, we can see that they have up to 25% drop of synthetic data (both DP and non-DP) compared to real data, even for their best GPT2-Large model.
>
> (Putta et al, 2023, https://openreview.net/pdf?id=LUql3ZOFwFD ) shows 8%-15% accuracy drop when comparing DP-synthetic data with eps=3 and real data without DP, see table 1. If we compare our and their results in the same settings (AGnews dataset, eps=3), our paper shows only 4.5% drop on DP-synthetic data compared to real data no-DP, additionally absolute value of accuracy of downstream model is 89.6% in our case vs 86.7% in their case.
>
> (Mattern et al., 2022, https://arxiv.org/pdf/2210.13918.pdf ) results are pretty close for real and synthetic data. Specifically they see about 2% drop of accuracy of downstream model when comparing real data and DP-synthetic data with eps=3 on IMDB dataset. However their chosen model for downstream classification seem to be suboptimal, specifically their performance on real IMDB data without DP is ~90.9%, while our experiments demonstrate 93.7% with no DP and t 90.6% +- 0.2% performance on DP synthetic data with strong epsilon = 3.
>
> *Summary*
>
> * (Yue et al, 2022) see significant drop of accuracy on one of their benchmarks.
> * (Putta et al, 2023) and (Mattern et al., 2022) both are using more complicated loss functions.
> * (Putta et al, 2023) results are generally worse than ours.
> * (Mattern et al., 2022) results look good in relative terms, but the absolute accuracy value is worse.
>
> Additionally, we want to point out that our approach updates only a small portion of LLMs parameters, and is much faster than tuning the whole model as was done in previous works mentioned above.
>
> We will expand our comparison section in 5.1 space permitting, in the final version if accepted.

---

### Meta-Review · Area_Chair_jDRc · 2023-12-11

**Metareview:**

(a) Summarize the scientific claims and findings of the paper based on your own reading and characterizations from the reviewers.

The authors propose privately finetuning LLMs to generate private texts foreach classes. The main difference compared to prior work is that (1) the pretraining data is deduplicated w.r.t. the downstream task, (2) long sequences are use as privacy units, and (3) prefix-LM loss sis used in fientuning. It is demonstrated that DP-synthetic data achieves better utility compared to DP-finetuning. It is demonstrated that LoRA improves upon finetuning under DP.

(b) What are the strengths of the paper?

The experiments are done thoroughly. It is a common practice to use public data pretrained models without checking whether the downstream private task has overlapping examples with the supposedly public data. This has been a widespread issue in the community as correctly emphasized in a seminal position paper "Considerations for differentially private learning with large-scale public pretraining" recently. The manuscript meticulously corrects this by running a deduplication on the public data with respect to the downstream private data of interest. While this is necessary for obvious reasons in privacy research, it is rarely done. Sometimes such a malpractice can jeopardize the whole paper. The item and effort spent to run such experiments is laudable. Another equally overlooked aspect of private training is hyper parameter tuning. It is almost a norm that one does not account for the multiple experiments done for hyper parameter tuning, in privacy research. This is a widespread practice that is theoretically not acceptable but practically overlooked. The proposed approach is, in some parts, motivated to combat such malpractice by the use of synthetic data. As private training is bypassed via private synthetic data, multiple runs do not accumulate the privacy leakage. Hyper parameter tuning can be done as before without the necessity of omitting all the intermediate training and the privacy leakage that happened. The scholarship of the authors in addressing these two widespread issues in privacy research is impressive. More practical privacy papers should follow this manuscript.

(c) What are the weaknesses of the paper? What might be missing in the submission?

The reviewers agree that the novelty of the proposed solution is not significant. Relatedly, the similarity to the prior work is not well explained. For example, Yue et al. (2022) show that DP synthetic yields good quality. It is clearly stated in the abstract of Yue et al. (2022): "Through extensive empirical analyses, we demonstrate that our method produces synthetic data that is competitive in terms of utility with its non-private counterpart" and across the paper. The difference between Yue et al. (2022)'s proposed solution and the solution proposed in this manuscript is subtle and perhaps not significant. Further, there has been several demonstrations of the gain in parameter efficient finetuning under privacy, such as  Yu, Da, et al. "Differentially private fine-tuning of language models." arXiv preprint arXiv:2110.06500 (2021).

**Justification For Why Not Higher Score:**

This paper shows well executed experiments but with little novel ideas. Given that the problem addressed is not new either and the motivation is also the same as prior work, it is difficult to justify the proposed method as a new approach. Some parts of the design is impressive, such as deduplication, but the idea that went into designing the proposed methods in this paper lacks novelty and depth.

**Justification For Why Not Lower Score:**

N/A

---

### Decision · Program_Chairs · 2024-01-16

Reject